# RNase H2, mutated in Aicardi-Goutières syndrome, promotes LINE-1 retrotransposition

Maria Benitez-Guijarro[1,†], Cesar Lopez-Ruiz[1,‡], Žygimantė Tarnauskaitė[2], Olga Murina[2], Mahwish Mian Mohammad[2,§], Thomas C Williams[2], Adeline Fluteau[2], Laura Sanchez[1], Raquel Vilar-Astasio[1], Marta Garcia-Canadas[1], David Cano[1], Marie-Jeanne HC Kempen[2], Antonio Sanchez-Pozo[3], Sara R Heras[1,3], Andrew P Jackson[2], Martin AM Reijns[2,*] [iD] & Jose L Garcia-Perez[1,2,**] [iD]

## Abstract

Long INterspersed Element class 1 (LINE-1) elements are a type of abundant retrotransposons active in mammalian genomes. An average human genome contains ~100 retrotransposition-competent LINE-1s, whose activity is influenced by the combined action of cellular repressors and activators. TREX1, SAMHD1 and ADAR1 are known LINE-1 repressors and when mutated cause the autoinflammatory disorder Aicardi-Goutières syndrome (AGS). Mutations in RNase H2 are the most common cause of AGS, and its activity was proposed to similarly control LINE-1 retrotransposition. It has therefore been suggested that increased LINE-1 activity may be the cause of aberrant innate immune activation in AGS. Here, we establish that, contrary to expectations, RNase H2 is required for efficient LINE-1 retrotransposition. As RNase H1 overexpression partially rescues the defect in RNase H2 null cells, we propose a model in which RNase H2 degrades the LINE-1 RNA after reverse transcription, allowing retrotransposition to be completed. This also explains how LINE-1 elements can retrotranspose efficiently without their own RNase H activity. Our findings appear to be at odds with LINE-1-derived nucleic acids driving autoinflammation in AGS.

**Keywords** Aicardi-Goutières Syndrome; LINE-1 retrotransposition; RNase H2; RNA:DNA hybrids
**Subject Categories** Chromatin, Epigenetics, Genomics & Functional Genomics; Immunology; Molecular Biology of Disease
The EMBO Journal (2018) 37: e98506

## Introduction

A significant proportion of most mammalian genomes consists of LINE-derived sequences, and the majority of these contain active LINE-1 (L1) elements. An average human genome contains more than 500,000 L1 copies but only 80–100 can currently mobilise (Brouha *et al*, 2003; Beck *et al*, 2010); these are termed retrotransposition-competent L1s or RC-L1s. As LINE-1 retrotransposition occurs randomly throughout the human genome, insertion events are sporadically mutagenic and can result in an array of genetic disorders [(Kazazian *et al*, 1988), reviewed in Ref. (Hancks & Kazazian, 2012; Garcia-Perez *et al*, 2016)]. RC-L1s are 6 kb in length, contain an internal promoter in their 5′-UnTranslated Region (UTR; Swergold, 1990), two intact open reading frames (coding for L1-ORF1p and L1-ORF2p) and end in a short 3′-UTR containing a variable sized poly-A tail required for retrotransposition (Scott *et al*, 1987; Doucet *et al*, 2015). L1-ORF1p is an RNA binding protein (Khazina & Weichenrieder, 2009) with nucleic acid chaperone activity (Martin & Bushman, 2001); L1-ORF2p codes for a protein with both endonuclease (EN, Feng *et al*, 1996) and reverse transcriptase activities (RT, Mathias *et al*, 1991; reviewed in Ref. Richardson *et al*, 2015). The enzymatic activities of L1-ORF1p and L1-ORF2p are strictly required for LINE-1 retrotransposition (Moran *et al*, 1996). In addition, L1-ORF2p contains a PCNA Interaction Protein motif (PIP) that allows interaction with PCNA (proliferating cell nuclear antigen) and is required for efficient L1 retrotransposition (Taylor *et al*, 2013).

LINE-1 retrotransposition occurs by a mechanism termed target primed reverse transcription (TPRT, Luan *et al*, 1993). Briefly, upon transcription and translation of a full-length L1 mRNA (Swergold, 1990; Alisch *et al*, 2006; Dmitriev *et al*, 2007), both L1-encoded proteins bind to their own L1 mRNA in a process termed

1 GENYO, Centro de Genómica e Investigación Oncológica: Pfizer - Universidad de Granada - Junta de Andalucía, PTS, Granada, Spain
2 MRC Human Genetics Unit, MRC, Institute of Genetics and Molecular Medicine, Western General Hospital, University of Edinburgh, Edinburgh, UK
3 Department of Biochemistry and Molecular Biology II, Faculty of Pharmacy, University of Granada, Granada, Spain
*Corresponding author. Tel: +44 131 651 8633; E-mail: martin.reijns@igmm.ed.ac.uk
**Corresponding author. Tel: +44 131 651 8717; E-mail: jose.garcia-perez@igmm.ed.ac.uk
†Present address: The University of Granada (Doctorate of Pharmacy Programme), Granada, Spain
‡Present address: OncoMark Ltd., Dublin, Ireland
§Present address: CNRS - UMR 8200, Institut Gustave Roussy, Villejuif, France

*cis*-preference (Wei *et al*, 2001), generating an L1 RiboNucleoprotein Particle (L1-RNP) that is considered a retrotransposition intermediate. Next, L1-RNPs access the nucleus by a process that does not require cell division (Kubo *et al*, 2006; Macia *et al*, 2017). In the nucleus, the L1 mRNA is reverse-transcribed and integrated in a new genomic location by TPRT (Luan *et al*, 1993; Cost *et al*, 2002; Piskareva *et al*, 2003; Piskareva & Schmatchenko, 2006). During TPRT, L1-ORF2p recognises and cleaves the bottom strand of genomic DNA in a consensus sequence (5′TTTT/AA and variants, Jurka, 1997), generating a free 3′OH that is used by the RT activity of L1-ORF2p to prime first-strand cDNA synthesis. As a result, an L1 RNA:cDNA hybrid covalently linked to the genome is generated. The mechanism of second-strand cDNA synthesis is not fully understood, but requires removal of the L1 mRNA from the generated hybrid, which allows second-strand cDNA synthesis to occur. In this context, there are two main classes of LINE elements: elements coding for a functional RNase H domain (mostly present in plants and lower eukaryotes) and elements that lack this domain (most mammals, including human LINE-1s; Malik *et al*, 1999; Olivares *et al*, 2002; Piskareva & Schmatchenko, 2006). RNase H enzyme activity degrades the RNA strand of RNA:DNA heteroduplexes, and how LINE elements without a functional RNase H domain achieve this is currently not known. Either way, it is assumed that upon removal of the LINE mRNA from the hybrid, the top strand of genomic DNA is cleaved, presumably by L1-ORF2p, releasing a 3′OH that is then used to prime second-strand cDNA synthesis, also presumed to be carried out by L1-ORF2p (Cost *et al*, 2002; Piskareva *et al*, 2003; Piskareva & Schmatchenko, 2006; reviewed in Richardson *et al*, 2015). The result of TPRT is the generation of a new LINE-1 insertion, usually flanked by short target site duplications (TSDs) and often 5′-truncated (Richardson *et al*, 2015).

Due to their mutagenic potential, the host has evolved a myriad of mechanisms to control LINE-1 expression and retrotransposition (recently reviewed in Ref. Garcia-Perez *et al*, 2016; Heras *et al*, 2014; Pizarro & Cristofari, 2016). Some repressors of L1 activity, when mutated, were shown to cause rare genetic disorders, such as Ataxia Telangiectasia (Coufal *et al*, 2011) and Aicardi-Goutières syndrome (AGS, reviewed in Ref. Volkman & Stetson, 2014). AGS is a type I interferonopathy where aberrant innate immune activation causes type I IFN production and clinical features reminiscent of a congenital viral infection of the brain (Crow & Manel, 2015). AGS is caused by mutations in one of seven genes: the 3′ exonuclease TREX1 (Crow *et al*, 2006a), the deoxynucleoside triphosphate triphosphohydrolase and putative ribonuclease SAM domain and HD domain 1 (SAMHD1; Rice *et al*, 2009), the adenosine deaminase acting on RNA 1 (ADAR1; Rice *et al*, 2012), the dsRNA cytosolic sensor IFN-induced helicase C domain containing protein 1 (IFIH1; Rice *et al*, 2014) and the three subunits of the ribonuclease H2 (RNase H2) endonuclease complex (RNASEH2A, RNASEH2B and RNASEH2C; Crow *et al*, 2006b; reviewed by Crow and Manel, 2015). In most cases, the IFN response in AGS patients is likely the consequence of the accumulation of aberrant cytoplasmic nucleic acids that activate innate immune receptors. Although the nature of such immunostimulatory nucleic acids has not been fully characterised, retrotransposons have been proposed as a possible source (Volkman & Stetson, 2014). In line with this, TREX1, SAMHD1 and ADAR1 have been shown to inhibit the mobility of LINE-1s (Stetson *et al*, 2008; Zhao *et al*, 2013; Orecchini *et al*, 2017), and RNase H2,

mutated in more than half of all known AGS patients (Crow *et al*, 2015), was suggested to similarly control LINE-1 retrotransposition (Volkman & Stetson, 2014).

Mammalian cells have two enzymes with the ability to degrade the RNA strand of RNA:DNA heteroduplexes: RNase H1 and RNase H2 (reviewed by Cerritelli and Crouch, 2009). RNase H1 has two isoforms: one localises to mitochondria and is essential for mitochondrial DNA replication (Cerritelli *et al*, 2003), and the second is nuclear and important for preventing R-loops and consequent transcription–replication conflicts (Nguyen *et al*, 2017; Parajuli *et al*, 2017; Shen *et al*, 2017). RNase H2, a heterotrimeric complex, is the predominant nuclear enzyme responsible for RNA:DNA hybrid degradation (reviewed by Reijns and Jackson, 2014), although this may depend on cell type. RNase H2 is essential for genome stability and also has the ability to cleave the 5′-phosphodiester bond of ribonucleotides embedded in a DNA duplex, an activity required for the removal of misincorporated ribonucleotides (Nick McElhinny *et al*, 2010; Hiller *et al*, 2012; Reijns *et al*, 2012) in a process called Ribonucleotide Excision Repair or RER (Sparks *et al*, 2012). Whereas the catalytic core of RNase H2 is present in the RNASEH2A subunit, all three subunits are required for its activity (Reijns & Jackson, 2014). The precise role for the accessory RNASEH2B and C subunits is not well understood, but a functional PCNA-interacting PIP motif is found in RNASEH2B (Chon *et al*, 2009), directing RNase H2 activity to replication and repair foci (Bubeck *et al*, 2011; Kind *et al*, 2014).

Here, we analyse the role of RNase H2 in LINE-1 retrotransposition and find that, in contrast to other AGS proteins, RNase H2 enzyme activity facilitates human LINE-1 retrotransposition. In the absence of RNase H2, overexpression of RNase H1 partially rescues the LINE-1 retrotransposition defect. Furthermore, overexpression of wild-type RNase H2 or a separation of function mutant of RNase H2 that can only cleave RNA:DNA heteroduplexes results in increased LINE-1 retrotransposition. The requirement for RNase H2 appears to be specific to retroelements without their own RNase H domain, suggesting that cellular RNase H activity is necessary to degrade the LINE-1 RNA in the RNA:cDNA hybrid generated during retrotransposition. Our findings call into question the existence of a unifying molecular mechanism for AGS pathogenesis centred around the accumulation of LINE-1 nucleic acids.

## Results

### LINE-1 retrotransposition is compromised in RNase H2 null HeLa cells

To test the effect of RNase H2 on LINE-1 retrotransposition, we generated a panel of clonal HeLa RNase H2 null cell lines using CRISPR/Cas9-mediated genome editing and guide RNAs (gRNAs) directed to the RNASEH2A subunit (Fig 1). Individual mutant clones were selected based on deletions/insertions observed after PCR amplification and sequencing of the target locus. Loss of RNASEH2A expression and reduced levels of RNASEH2B and C were shown by Western analysis in knockout clones (KO, Fig 1A). Additionally, loss of RNase H activity in cell lysates against single-embedded ribonucleotides was confirmed using a FRET-based fluorescent substrate release assay (Fig 1B). Finally, increased fragmentation of

genomic DNA from KO clones after RNase H2 treatment and alkaline gel electrophoresis indicated the presence of large numbers of embedded ribonucleotides in KO clones (Fig 1C), a well-known consequence of RNase H2 deficiency (Nick McElhinny *et al*, 2010; Hiller *et al*, 2012; Reijns *et al*, 2012). For these and subsequent experiments, KO clones were compared to parental cells as well as CRISPR control clones (C) that retained significant RNase H2 activity.

These RNase H2 null and control clones were used in a cell-based LINE-1 mobilisation assay, which makes use of an active human LINE-1 element (L1.3, Sassaman *et al*, 1997) tagged with a reporter gene that can only be activated after a round of retrotransposition (Figs 1D and EV1A, Moran *et al*, 1996). As a reporter gene, we used *mblastI* (Morrish *et al*, 2002; Goodier *et al*, 2007), which activates the blasticidin-resistant gene after retrotransposition (Fig 1D, plasmid JJ101/L1.3). Importantly, this assay is quantitative, and the resulting number of drug-resistant colonies provides a readout of retrotransposition activity (Morrish *et al*, 2002; Goodier *et al*, 2007). Surprisingly, LINE-1 retrotransposition was severely reduced in the RNase H2 null lines ($n = 6$ clones), with an average level of retrotransposition of $6.0 \pm 2.6\%$ (mean $\pm$ SD) compared to control lines ($n = 6$, $94 \pm 22\%$; $P = 0.0022$) or parental cells (set to 100%; Fig 1E and F). As expected, control plasmids containing RT mutated LINE-1 (JJ101/L1.3-D702A) failed to retrotranspose in all cell lines (Fig 1F), whereas similar numbers of blasticidin-resistant colonies were generated for all cell lines after transfection with a control vector (pcDNA6.1, expressing the blasticidin-resistant gene, Fig EV1B). Retrotransposition of an EN-mutant LINE-1 (JJ101/L1.3-D205A) was similarly low in wild type and null cells, suggesting that DNA lesions in null cells (Reijns *et al*, 2012) are not used as integration sites by endonuclease mutant LINE-1s (Fig 1F).

To confirm these data, we next used an L1 retrotransposition assay that activates expression of luciferase upon retrotransposition (Fig EV1C). This assay has two advantages over the drug resistance-based assay: (i) it does not require selection with antibiotic or generation of drug-resistant colonies, and (ii) retrotransposition levels are measured 96 h post-transfection. This way long-term culturing is avoided, and possible confounding effects due to differential growth between controls and RNASEH2A-KO cell lines can therefore be ruled out. Using this luciferase-based assay, we observed a 63% reduction in L1 retrotransposition in RNase H2 null cell lines ($n = 3$, $31 \pm 6.2\%$) compared to controls ($n = 3$, $84 \pm 8.3\%$; $P = 0.0009$; Fig EV1D). Furthermore, to exclude an indirect effect of RNase H2 deficiency on L1 expression, we performed Western blot analysis using an antibody against endogenous L1-ORF1p. Although there was some variation between clones, we confirmed that L1-ORF1p is expressed at similar levels in RNASEH2A-KO cells and parental cells (Fig EV1E–G). In summary, these data suggest that RNase H2 promotes LINE-1 retrotransposition in cultured HeLa cells.

## LINE-1 retrotransposition is compromised in RNase H2 null U2OS and HCT116 cells

To determine whether RNase H2 activity is required for LINE-1 mobilisation in other cellular backgrounds, we next analysed retrotransposition in colon carcinoma (HCT116) and osteosarcoma (U2OS) cells. Firstly, using the same CRISPR/Cas9 strategy, we generated RNASEH2A-KO and control clones using HCT116 p53$^{-/-}$ cells. Western blotting, enzyme activity assays and analysis of genomic ribonucleotide incorporation confirmed absence of RNase H2 activity in two clonal knockout cell lines (KO, Appendix Fig S1A–C). Using the JJ101/L1.3-based assay, we observed significantly reduced retrotransposition in HCT116 RNASEH2A-KO clones ($n = 2$, $29 \pm 8.5\%$) compared to controls cells ($n = 4$, $100 \pm 14\%$, $P = 0.0028$; Appendix Fig S1D and E).

Secondly, we generated clonal RNASEH2A-KO and control lines using U2OS cells (Fig 2A–C), which were assayed for retrotransposition alongside parental cells. Notably, and consistent with our data for HeLa and HCT116 p53$^{-/-}$ RNase H2 null cells, we observed a substantial reduction in LINE-1 retrotransposition in the RNASEH2A-KO lines ($n = 2$, $25 \pm 9.5\%$) compared to parental cells (set to 100%) and a wild-type control clone ($n = 2$, $94 \pm 8.2\%$; $P = 0.016$; Fig 2D). Additional controls showed that all clones generated similar number of blasticidin-resistant colonies upon transfection with the control vector pcDNA6.1 and that an L1 RT-mutant construct (L1.3-D702A) failed to retrotranspose in all U2OS clones tested (Fig 2D).

In summary, LINE-1 retrotransposition is strongly reduced in multiple RNase H2 null clones of three different cell lines. We therefore conclude that cellular RNase H2 activity is required for efficient LINE-1 retrotransposition.

---

**Figure 1.  Reduced LINE-1 retrotransposition in RNase H2 null HeLa cells.**

A   Western blot analysis shows absence of RNASEH2A and reduced RNASEH2B and C in RNASEH2A-KO clones (KO1-6), compared to parental cells or control clones (C1-5). Tubulin was used as a loading control.

B   RNase H assay shows absence of activity against single-embedded ribonucleotides in KO clones, with a smaller, but consistent reduction in all control clones. Activity in parental HeLa cells set at 100%. Data points represent the mean of three technical replicates for individual clones. Lines indicate the mean of six biological replicates (C1-6 and KO1-6) $\pm$ SEM.

C   High levels of genome-embedded ribonucleotides in KO clones. Genomic DNA isolated from parental cells, KO and control clones, was RNase H2 treated and separated by alkaline gel electrophoresis. Smaller fragments indicate larger numbers of embedded ribonucleotides.

D   Schematic of retrotransposition vector JJ101/L1.3 (see also Fig EV1A). Within L1-ORF2p, relative positions of EN (endonuclease), RT (reverse transcriptase) and C (cysteine-rich) domains are indicated. Orange box with backward BLAST label depicts the retrotransposition indicator cassette *mblastI*.

E   Quantification of L1-WT retrotransposition, normalised to the level in parental cells and normalised for transfection efficiency (TE), set to 100% for comparison. Data points represent the mean of three technical replicates for individual clones. Lines indicate the mean of six biological replicates (C1-6 and KO1-6) $\pm$ SEM (representative of six independent experiments). Mann–Whitney test; **$P < 0.001$

F   Representative retrotransposition assay conducted in parental cells, control clones (C1-6) and RNASEH2A-KO clones (KO1-6). Cells were transfected with JJ101/L1.3-derived vectors containing an active human LINE-1 (WT-hL1, element L1.3), an EN-mutant LINE-1 (ENm-hL1, L1.3 D205A) or an RT-mutant LINE-1 (RTm-hL1, L1.3 D702A).

Source data are available online for this figure.

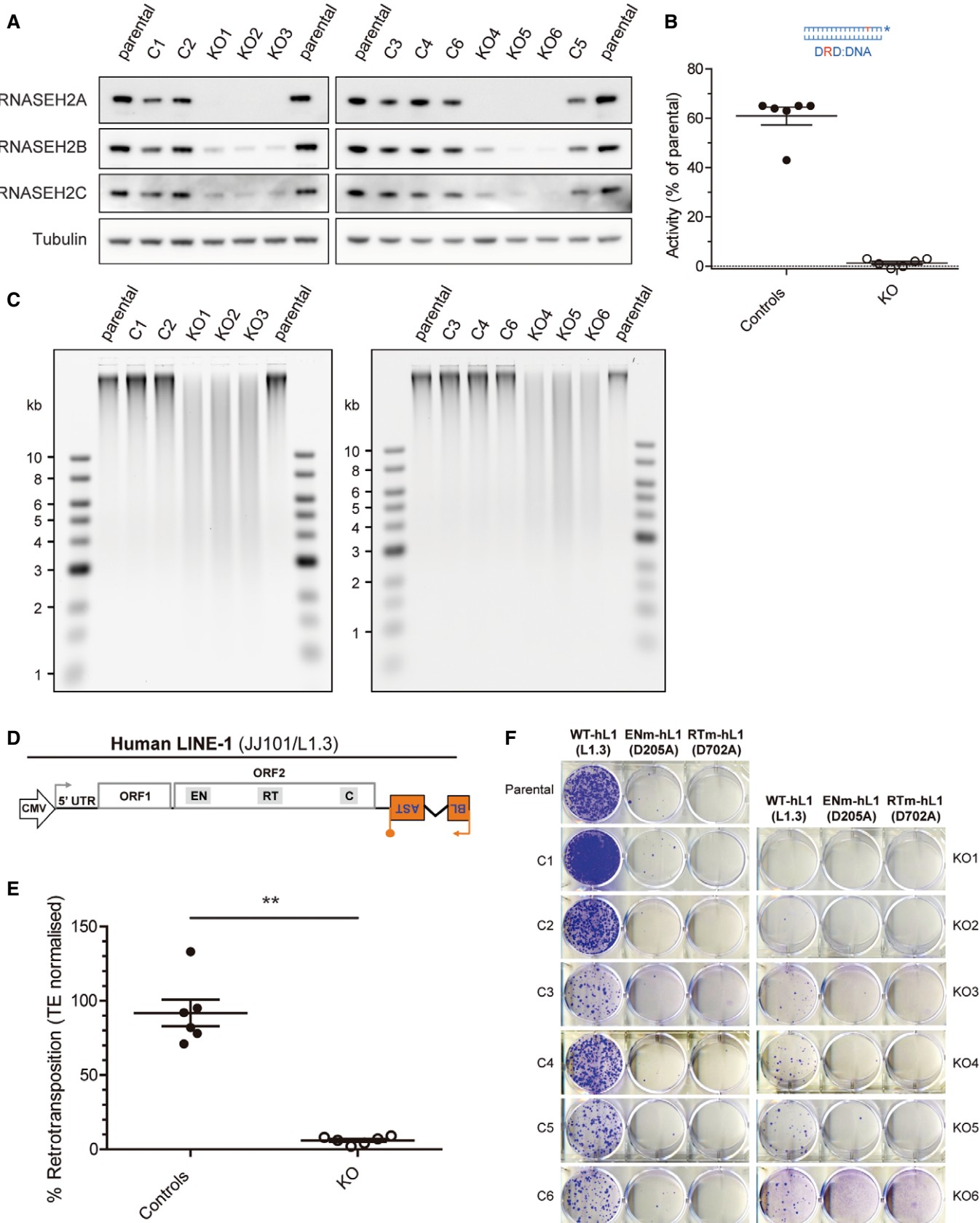

**Figure 1.**

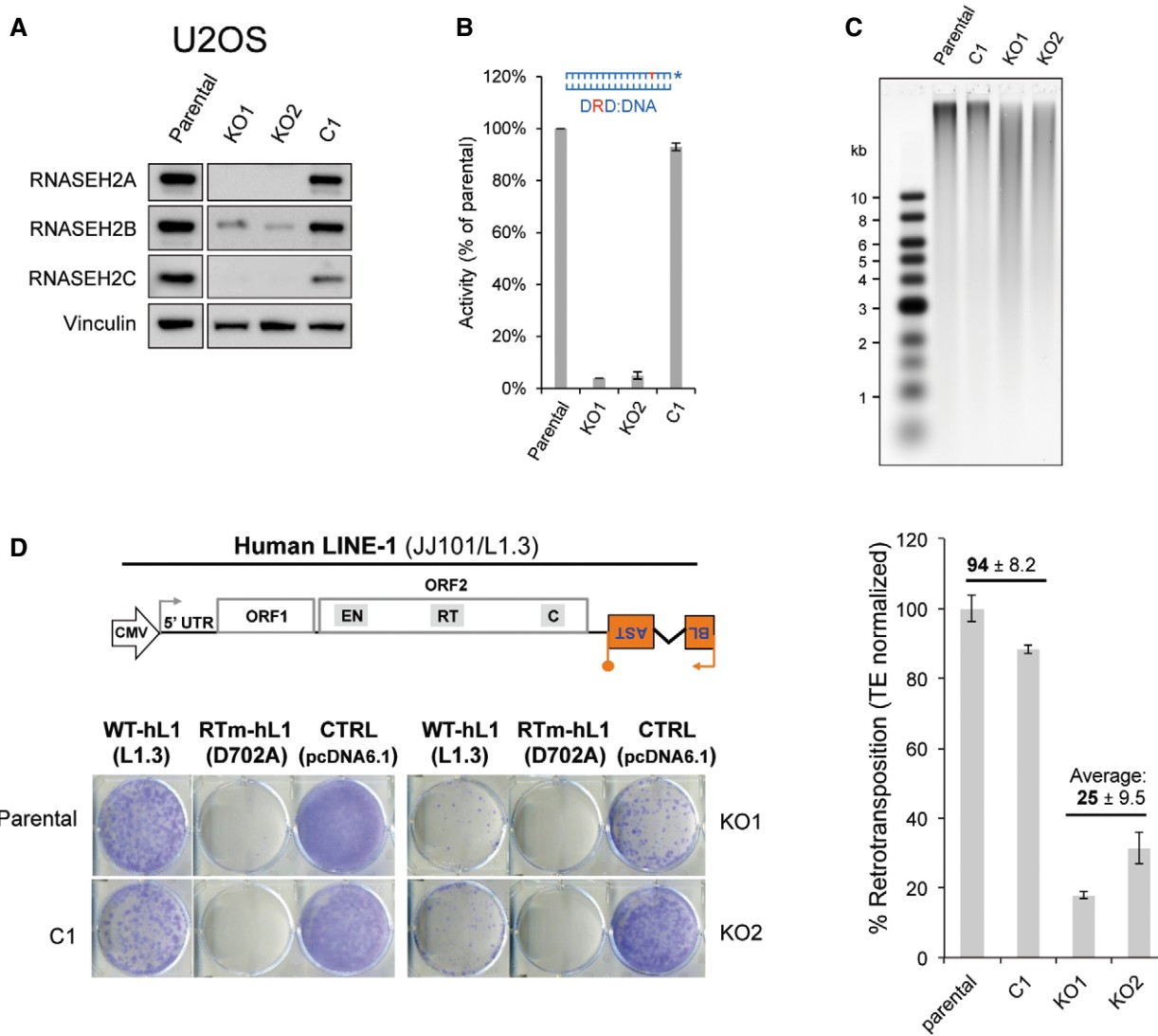

**Figure 2.   Reduced LINE-1 retrotransposition in RNase H2 null U2OS cells.**

A   Western blot analysis shows absence of RNASEH2A and reduced RNASEH2B and C in RNASEH2A-KO clones (KO1, KO2), compared to parental cells or a control clone (C1). Vinculin was used as a loading control.

B   RNase H assay shows absence of activity against single-embedded ribonucleotides in KO clones, compared to control cells. Activity in parental U2OS cells set at 100%. Mean ± SD for two independent experiments.

C   High levels of genome-embedded ribonucleotides in U2OS RNASEH2-KO clones. Genomic DNA was isolated from parental cells, KO and control clones, RNase H2 treated and separated by alkaline gel electrophoresis. Smaller fragments indicate more genome-embedded ribonucleotides.

D   Schematic of plasmid JJ101/L1.3 and representative retrotransposition and toxicity assays conducted in parental U2OS cells, a control clone (C1), and two RNASEH2A-KO clones (KO1 and KO2). Cells were transfected with vectors containing an active human LINE-1 (WT-hL1, element L1.3), an RT-mutant (RTm-hL1, L1.3 D702A), or a toxicity control vector (CTRL, pcDNA6.1). Quantification of L1-WT retrotransposition, with the level in parental cells set to 100% for comparison. Plotted, mean ± SD for three technical replicates. Numbers indicate the average ± SD of *n* = 2 controls (parental, C1) and *n* = 2 (KO1, KO2) (representative of three independent experiments).

Source data are available online for this figure.

## RNase H2 facilitates mobilisation of non-LTR retroelements but is dispensable for LTR-retrotransposons and DNA-transposons

During LINE-1 insertion by target primed reverse transcription (TPRT), after endonucleolytic cleavage of genomic DNA and first-strand cDNA synthesis by L1-ORF2p, an L1 RNA:cDNA hybrid attached to the genome is generated. The RNA in the hybrid must be removed prior to starting second-strand cDNA synthesis. Because human L1-ORF2p lacks RNase H activity (Mathias *et al*, 1991; Malik *et al*, 1999; Cost *et al*, 2002; Piskareva *et al*, 2003; Piskareva & Schmatchenko, 2006), we reasoned that cellular RNase H2 may instead degrade the RNA in the L1 RNA:cDNA hybrid. To test whether RNase H2 performs this function more generally for LINE elements lacking an RNase H domain, we next investigated the impact of RNase H2 deficiency on retrotransposition of a LINE-2 element from zebrafish (ZfL2-2). Notably, ZfL2-2 does not contain

an RNase H domain and is active in human cells lines (Sugano et al, 2006; Garcia-Perez et al, 2010). We tagged this LINE-2 element with an mneoI retrotransposition indicator cassette, which confers resistance to neomycin/G418 upon retrotransposition (Freeman et al, 1994; Moran et al, 1996; Fig 3A), and measured activity in the HeLa RNASEH2A-KO and control clones. An active human LINE-1 tagged with the same retrotransposition indicator cassette (vector JM101/L1.3, Fig 3A) was used in parallel as a control. The JM101/L1.3-mneoI vector produced results very similar to JJ101/L1.3-mblastI (Figs 1 and 3C). Consistent with our hypothesis, ZfL2-2-mneoI retrotransposition was significantly reduced in null clones ($n = 5$, $1.6 \pm 0.91\%$) compared to parental cells (set to 100%) and control clones ($n = 5$, $85 \pm 9.5\%$; $P = 0.0079$; Figs 3A–C and EV2A). Notably, when ZfL2-2 was assayed in the U2OS RNASEH2A-KO and control clones, we observed a virtually identical outcome (Fig EV2B; $n = 2$ controls, $105 \pm 3.5\%$ vs. $n = 2$ KO, $14 \pm 2.4\%$; $P = 0.0012$). As expected, we detected a similar reduction in the retrotransposition rate of a human LINE-1 element in both HeLa and U2OS RNASEH2A-KO clones using the mneoI-based assay (Figs 3A–C and EV2). Taken together, these data suggest that LINE elements lacking a functional RNase H domain rely on cellular RNase H2 activity to retrotranspose efficiently. This would make RNase H2 an integral part of the LINE retrotransposition machinery.

In line with this model, we reasoned that retrotransposons that do contain a functional RNase H domain would not rely on cellular RNase H2 activity to retrotranspose efficiently. Active LTR-retrotransposons from the mouse genome contain a functional RNase H domain within their pol gene (Doolittle et al, 1989); they generate cytoplasmic RNA:cDNA hybrid intermediates during their retrotransposition cycle, with RNA degradation required to complete LTR-retrotransposition. We therefore measured retrotransposition of MusD, an active LTR-retrotransposon from the mouse genome (Doolittle et al, 1989; Mager & Freeman, 2000). To do so, we tagged an active MusD element with the neo<sup>TNF</sup> retrotransposition indicator cassette (pCMVMusD-6 neoTNF; Fig 3D, Appendix Fig S2A; Ribet et al, 2004) and tested its activity in RNASEH2A-KO and control clones. Consistent with our model, the level of MusD retrotransposition was similar in HeLa control and RNASEH2A-KO clones (Fig 3D, Appendix Fig S2B). Similarly, no difference in MusD activity was observed either when comparing U2OS RNase H2 wild-type and null cells (Appendix Fig S2C). Thus, these data strongly suggest that

retrotransposons containing a functional RNase H domain do not depend on cellular RNase H2 activity for their mobilisation.

We next tested whether mobilisation of DNA-transposons, that move by a cut-and-paste mechanism that does not involve reverse transcription, would require cellular RNase H2 activity. We employed an active Tc1-like resurrected DNA-transposon termed Sleeping Beauty (SB; Ivics et al, 1997) that transposes very efficiently in human cells. Using an SB transposition assay based on G418 selection, no differences in transposition rates were observed when comparing RNase H2 null clones and controls, for either HeLa (Fig 3E) or U2OS (Appendix Fig S2D) cell lines.

We therefore conclude that neither LTR-retrotransposons that code for RNase H activity, nor DNA-transposons, rely on cellular RNase H2 activity for their mobilisation.

### RNase H2 overexpression increases LINE-1 retrotransposition

If RNase H2 is directly required for LINE-1 retrotransposition, it may be expected that overexpression of RNase H2 might further increase retrotransposition efficiency, and we set out to test this. RNase H2 is a heterotrimeric enzyme, and overexpression of the catalytic subunit alone does not significantly increase cellular activity (KR Astell, MAM Reijns & AP Jackson, unpublished data). We therefore co-transfected HeLa and U2OS cells with three plasmids each expressing one of the RNase H2 subunits tagged with a V5 epitope tag and an engineered human LINE-1 construct tagged with the mblastI indicator cassette (JJ101/L1.3). As controls, we transfected cells with a β-arrestin expression vector, a negative control (−ve) that does not significantly affect L1 retrotransposition (Bogerd et al, 2006), or with an APOBEC3A overexpression vector, a positive control (+ve) that strongly inhibits LINE-1 retrotransposition (Bogerd et al, 2006; Richardson et al, 2014). Cells were also co-transfected in a parallel assay with the control vector pcDNA6.1 and resulting colony numbers used for normalisation to control for potential toxic side effects of cDNA overexpression. Notably, when HeLa or U2OS cells were co-transfected with equal amounts of overexpression plasmids for each RNase H2 subunit (ratio 1:1:1) and vector JJ101/L1.3, we detected a significant increase in retrotransposition when compared to the β-arrestin control (2.1-fold in HeLa and 1.7-fold in U2OS, Fig 4). In agreement with previous reports (Bogerd et al, 2006), overexpression of APOBEC3A reduced LINE-1

---

**Figure 3. RNase H2 activity is required for LINE activity but dispensable for LTR-retroelement and DNA-transposon activity.** ▶

A Schematic of retrotransposition vectors Zfl2-2mneoI and JM101/L1.3. The relative position of the EN domain (endonuclease), RT domain (reverse transcriptase) and C domain (cysteine-rich), if present, is indicated. The purple box with a backward NEO label depicts the retrotransposition indicator cassette mneoI.

B Representative retrotransposition assays conducted in parental cells, control (C2) and RNASEH2A-KO (KO2) clones. Cells were transfected with vectors containing an active human LINE-1 (WT-hL1, element L1.3), an RT-mutant human LINE-1 (RTm-hL1, L1.3 D702A), or an active zebrafish LINE-2 (WT-zL2, element Zfl2-2).

C Quantification of WT-hL1 (circles) and WT-zL2 (squares) retrotransposition in HeLa cells, normalised to the level in parental cells (set at 100%). Data points represent the mean of three technical replicates for individual clones. Lines indicate the mean of five biological replicates (C2-6 and KO2-6) $\pm$ SEM (representative of three independent experiments). For WT-hL1, in control lines ($n = 5$) retrotransposition levels averaged $83 \pm 2.5\%$; in null lines ($n = 5$) retrotransposition levels averaged $7 \pm 2.3\%$. Mann–Whitney test; \*\*$P < 0.001$.

D Left, schematic of a neo<sup>TNF</sup> tagged MusD LTR-retrotransposon. The relative position of the gag, pro and pol genes is indicated. The purple box with a backward NEO label depicts the retrotransposition indicator cassette neo<sup>TNF</sup>. Right, quantification of MusD retrotransposition, normalised to the level in parental cells (set at 100%). Data points represent the mean of three technical replicates for individual clones. Lines indicate the mean of six biological replicates (C1-6 and KO1-6) $\pm$ SEM (representative of three independent experiments). t-test; ns, $P > 0.05$.

E Left, schematic of the two plasmids used in Sleeping Beauty transposition assays. The purple box with a NEO label depicts the neo expression cassette, flanked by Terminal Inverted Repeats (TIR). Underneath, a representative result is shown for DNA-transposition assays (pT2neo + SB100x) or controls (only pT2neo). Right, quantification of the SB transposition results (pT2neo + SB100x samples), with the level in parental cells set at 100% for comparison. Mean $\pm$ SD for $n = 3$ technical replicates (representative of three independent experiments).

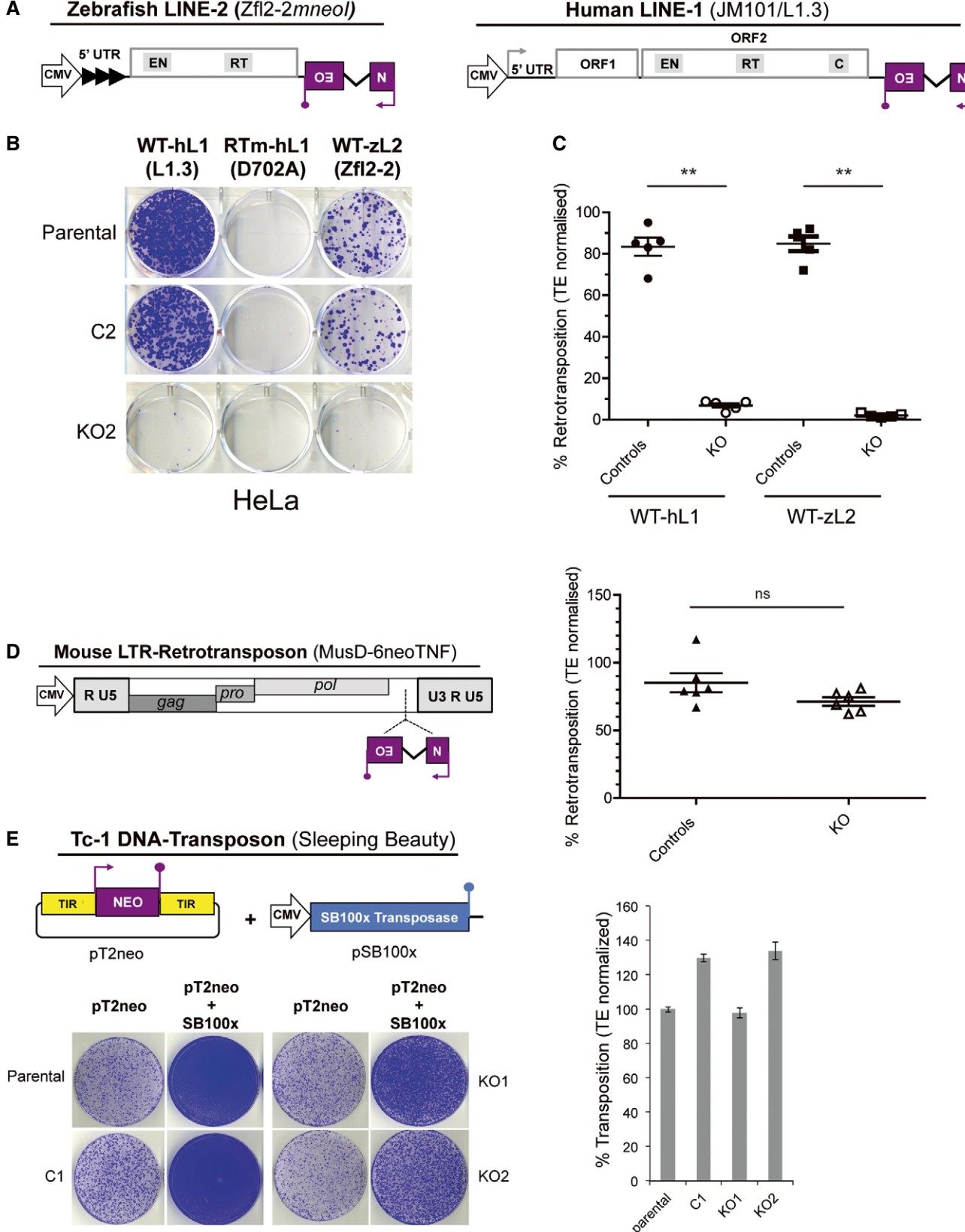

**Figure 3.**

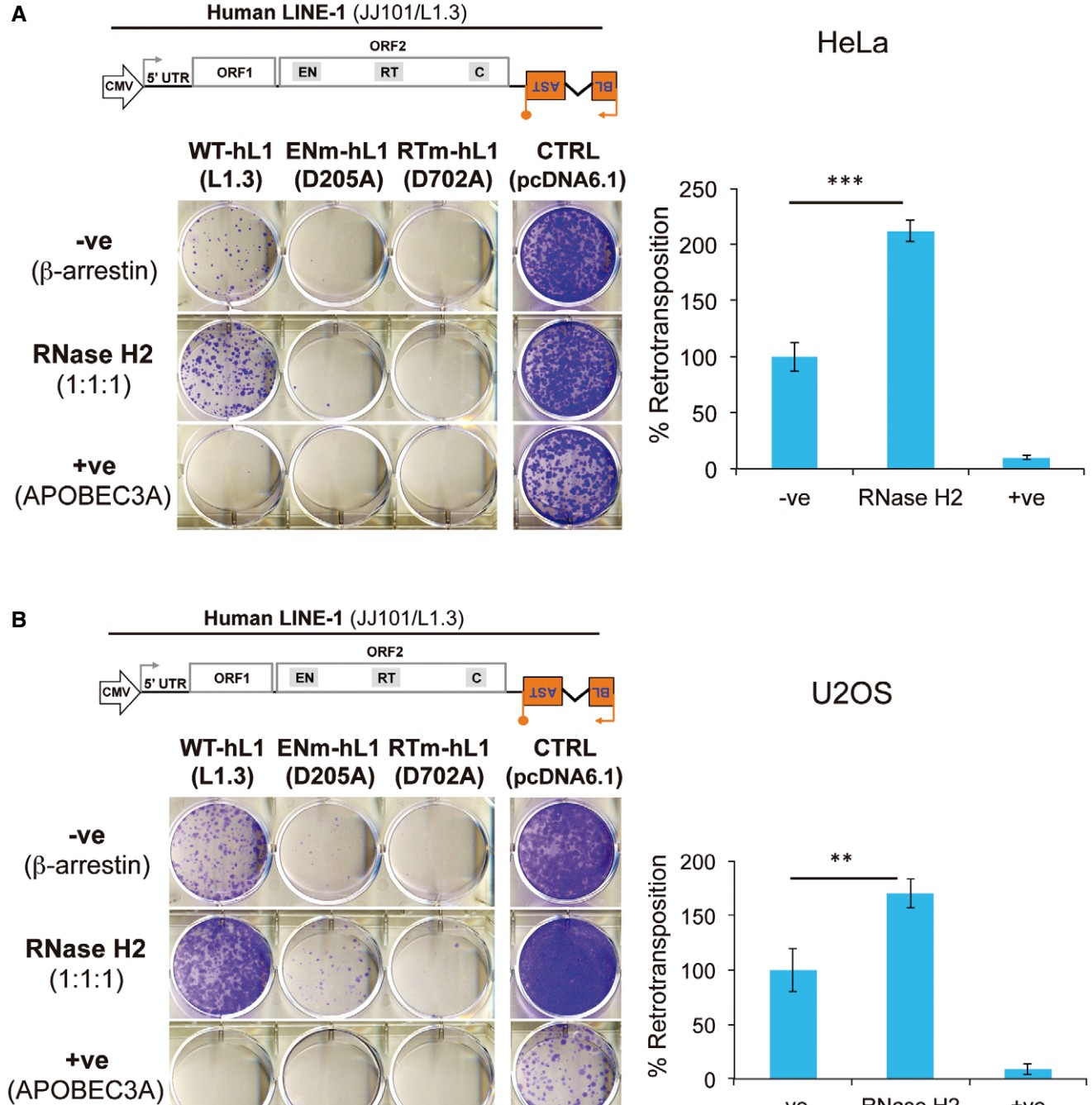

**Figure 4.  RNase H2 overexpression increases LINE-1 retrotransposition in HeLa and U2OS cells.**

A, B   Panels (A) (HeLa cells) and (B) (U2OS cells) follow the same nomenclature. Shown is a representative result for retrotransposition and toxicity assays, underneath a schematic of the retrotransposition vector JJ101/L1.3 used. Cells were transfected with JJ101/L1.3-based vectors [as indicated: WT-hL1 (L1.3), active human LINE-1; ENm-hL1 (D205A), EN-mutant; RTm-hL1 (D702A), RT-mutant] or the toxicity control vector (CTRL, pcDNA 6.1), alongside an expression vector for β-arrestin as a negative control (−ve), the three RNase H2 subunits (RNase H2, at a 1:1:1 ratio), or a plasmid expressing APOBEC3A as a positive control (+ve) known to restrict LINE-1 retrotransposition. Right panel, quantification of this retrotransposition assay; with the level in cells co-transfected with β-arrestin set at 100% for comparison. Values were normalised for transfection efficiency and toxicity. Mean ± SD for $n$ = 3 technical replicates (representative of four independent experiments). Unpaired two-sided $t$-test; **$P$ < 0.01; ***$P$ < 0.001.

retrotransposition to ~10% of control levels (Fig 4). As expected, RT-mutant LINE-1s failed to retrotranspose in any condition tested and the retrotransposition of EN-mutant LINE-1s was not affected

by RNase H2 overexpression (Fig 4), suggesting that RNase H2 cleavage at misincorporated ribonucleotides does not provide an entry point for endonuclease mutant LINE-1s. Western blot analysis

showed that the three V5-tagged RNase H2 subunits were expressed, although they were not detected at the same level (Fig EV3A). Through optimisation, we found that a 14:7:1 transfection ratio (for RNASEH2A, B and C, respectively) produced the most similar expression levels for each of the subunits (Fig EV3A). When measuring retrotransposition at this 14:7:1 ratio, LINE-1 retrotransposition increased again, this time by ~1.7-fold for HeLa cells and ~1.4-fold for U2OS cells (Fig EV3B and C). Thus, rather than reduced LINE-1 retrotransposition being an indirect effect of RNase H2 deficiency, these data are consistent with a direct role for cellular RNase H2 in facilitating LINE-1 retrotransposition.

**Complementation of KO cells with wild-type, but not separation of function RNASEH2A rescues LINE-1 retrotransposition**

RNase H2 can cleave at ribonucleotides embedded in double-stranded DNA and can hydrolyse the RNA strand of RNA:DNA heteroduplexes, raising the question as to which of these two activities is required to promote LINE-1 retrotransposition. Elegant work in yeast identified two amino acid changes in the RNase H2 catalytic subunit that severely abrogates its activity against single-embedded ribonucleotides, while retaining activity against RNA:DNA heteroduplexes (Chon *et al*, 2013). We took advantage of this "separation of function" (SoF) mutant to try to address this question (Fig EV4A). First, we tested whether the biochemical characteristics of recombinant human RNase H2 with the equivalent mutations (P40D/Y210A) were similar to that of the yeast enzyme (Fig 5A–D, and Chon *et al*, 2013). As expected, recombinant human SoF RNase H2 has virtually no activity against single-embedded ribonucleotides (Fig 5A and B), but retains activity against RNA:DNA heteroduplexes (Fig 5C and D). However, activity against RNA:DNA heteroduplex substrate was reduced compared to wild-type RNase H2, and unexpectedly, we also found that the SoF mutant produced longer RNA products when compared to wild-type RNase H2 (Fig 5C), suggestive of an altered cleavage pattern. This difference in cleavage was observed even with higher enzyme concentrations and at longer incubation times (Fig EV4B).

We next complemented a HeLa RNASEH2A-KO clone (KO1) using retroviral vectors to generate stable cell lines expressing either wild-type or SoF RNASEH2A. Cell lines expressing the empty vector (EV) or catalytically inactive (catalytic dead, CD) RNASEH2A (D34A/D169A; Reijns *et al*, 2011) were also generated and used as controls. Western blotting confirmed expression of RNASEH2A and the consequent stabilisation of RNASEH2B and C in complemented cells at levels indistinguishable from control cells (Fig 5E). As expected, complementation with WT RNASEH2A but not EV, SoF or CD reduced the level of ribonucleotide incorporation to a level similar to that observed in wild-type controls (Fig EV4C). This was consistent with the level of RNase H activity against single-embedded ribonucleotides measured in cell lysates in complemented cells (Figs 5F and EV4D and E). On the other hand, activity against RNA:DNA heteroduplexes was similar for the WT complemented cells (+WT) and wild-type control cells (C1), whereas the SoF complemented cells displayed < 50% activity (Figs 5G and EV4F and G), in line with the observed reduction in activity of recombinant SoF RNase H2. RNase H1 is expressed in all of these cells, explaining the residual activity against RNA:DNA substrate in RNASEH2A-KO cells complemented with empty vector or RNASEH2A-CD. Notably, the altered cleavage pattern on RNA:DNA hybrids was also detected in cell lysates from RNASEH2A-SoF complemented cells (Fig EV4G). These data suggest that, although the P40D/Y210A amino acid changes in human RNase H2 do act as separation of function mutations, RNase H activity against RNA:DNA heteroduplexes of the SoF mutant is compromised both *in vitro* and *in vivo*.

Retrotransposition assays on the complemented cell lines using vector JJ101/L1.3 demonstrated that complementation with wild-type RNASEH2A allowed efficient retrotransposition (Fig 5H). A second HeLa RNASEH2A-KO clone (KO2) complemented with WT RNASEH2A also showed rescue of the L1-retrotransposition defect (Fig EV4H), confirming that reduced retrotransposition in RNASEH2A-KO cells is due to lack of RNase H2 activity and not due to potential off-target effects of CRISPR/Cas9. In contrast, complementation with RNASEH2A-SoF failed to rescue the L1 retrotransposition defect, with levels similar to those seen in EV or CD

▶

**Figure 5.    Complementation of RNASEH2A-KO cells with wild-type, but not separation of function RNASEH2A rescues LINE-1 retrotransposition.**

A, B    RNase H activity assays against single-embedded ribonucleotides using recombinant purified proteins (WT-RNase H2 and SoF-RNase H2). Note that only WT-RNase H2 shows activity in this assay. Plotted, mean ± SEM for three independent experiments.

C, D    RNase H activity assays against RNA:DNA heteroduplexes using recombinant purified proteins (WT-RNase H2 and SoF-RNase H2). Note that the pattern of products generated by SoF-RNase H2 is different from the wild-type pattern. Plotted, mean ± SEM for three independent experiments.

E    Western blot analysis of RNase H2 expression in RNASEH2A-KO HeLa cells complemented with the indicated retroviral vector (EV, empty vector; WT, wild-type RNASEH2A; SoF, RNASEH2A-P40D/Y210A; CD, RNASEH2A-D34A/D169A, see main text for details). Tubulin was used as a loading control.

F    RNase H activity against single-embedded ribonucleotides in RNASEH2A-KO cells is only rescued by wild-type RNASEH2A (KO1 + WT). DRD:DNA heteroduplex (18 bp; ribonucleotide-containing strand 3′-labelled) was incubated with increasing amounts of whole-cell lysate from the indicated cell line and separated by denaturing PAGE. The graph shows mean values ± SEM for three independent experiments.

G    RNase H activity against RNA:DNA heteroduplexes in RNASEH2A-KO cells is rescued by wild-type (KO1 + WT), not by CD RNASEH2A (KO1 + CD) or the empty vector (KO1 + EV). Note reduced activity and the difference in cleavage pattern produced by SoF RNASEH2A (KO1 + SoF). RNA:DNA heteroduplex (18 bp; RNA strand 3′-labelled) was incubated with increasing amounts of whole-cell lysate from the indicated cell line and separated by denaturing PAGE. Plotted, mean ± SEM for three independent experiments.

H    Only wild-type RNASEH2A rescues the LINE-1 retrotransposition defect in RNASEH2A-KO cells. Left, representative retrotransposition and toxicity assays conducted in the four complemented lines. Cells were transfected with vectors containing an active human LINE-1 (WT-hL1, L1.3), an RT-mutant LINE-1 (RTm-hL1, D702A) or a toxicity control plasmid (CTRL, pcDNA 6.1). Right, quantification of L1-WT retrotransposition. For comparison, the retrotransposition level in KO1 cells complemented with the empty vector (EV) was set at 100%. Mean ± SD for *n* = 3 technical replicates (representative of six independent experiments).

Source data are available online for this figure.

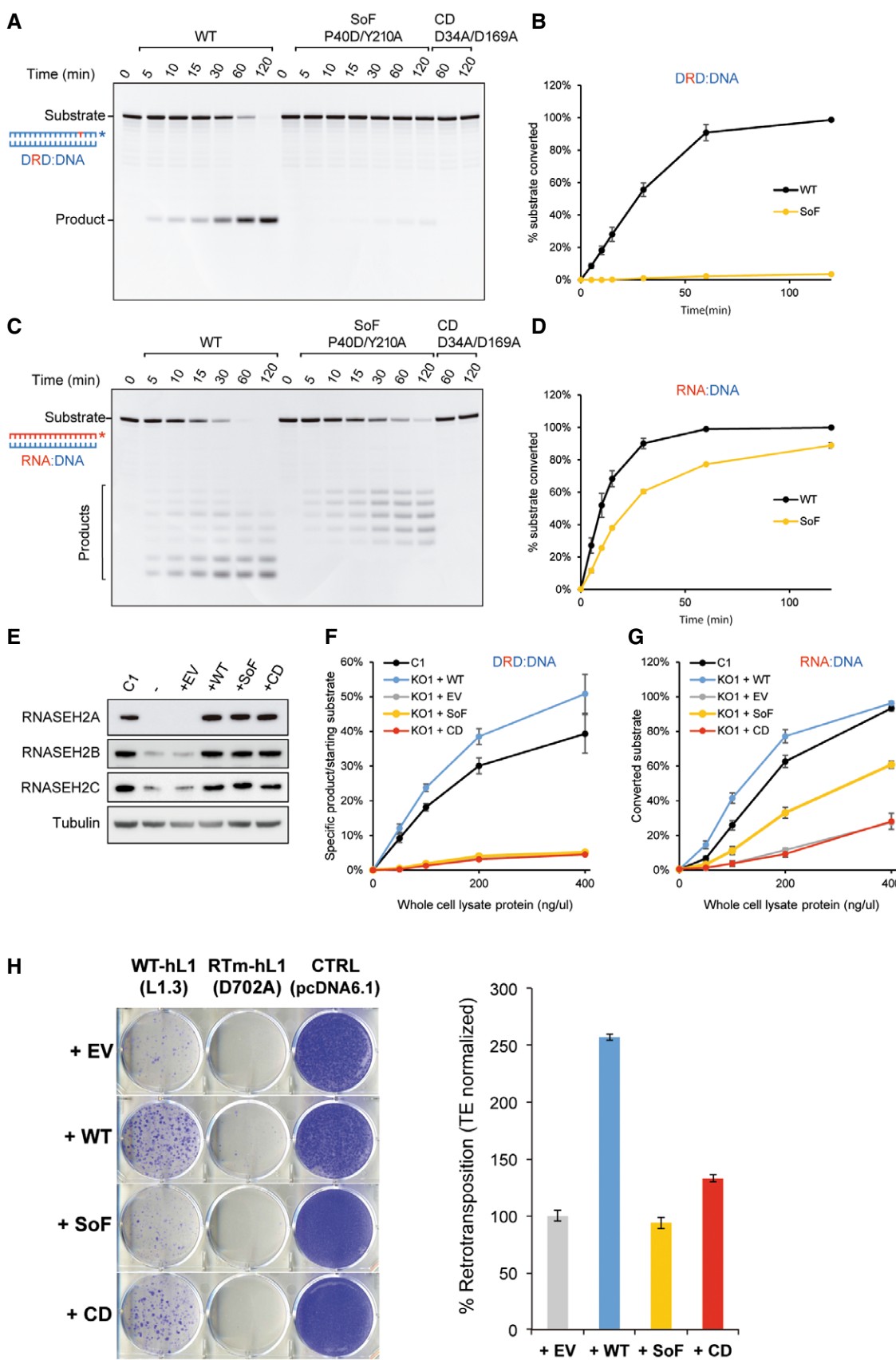

Figure 5.

complemented cells (Fig 5H). Failure to complement the LINE-1 retrotransposition defect in RNase H2 null cells with the RNASEH2A separation of function mutant may be a direct consequence of its altered biochemical characteristics, resulting in its failure to fully/efficiently degrade the RNA in the LINE-1 RNA:cDNA hybrid. However, it is formally possible that RNase H2 promotes LINE-1 retrotransposition through its activity on embedded ribonucleotides.

### No increased mutation rates in newly inserted LINE-1 elements in RNase H2 null cells

It has been shown that during human immunodeficiency virus type 1 (HIV-1) reverse transcription, ribonucleotides are misincorporated at high frequency by the RT of HIV-1, especially in macrophages (Kennedy *et al*, 2012). Notably, RNase H2 was shown in a high-throughput screen to be important for HIV-1 infection (Genovesio *et al*, 2011), and it may be involved in removing such embedded ribonucleotides. Although drastic differences exist among the mechanism of retroviral insertion and LINE-1 retrotransposition, it is therefore possible that ribonucleotides are misincorporated during L1 reverse transcription and/or second-strand synthesis and that their removal by RNase H2-dependent RER is important to allow efficient retrotransposition to occur. Ribonucleotide misincorporation in RNase H2-deficient yeast was shown to cause high rates of Top1-dependent 2–5-bp deletions (Nick McElhinny *et al*, 2010; Kim *et al*, 2011). These mutations are most likely to occur at tandem dinucleotide repeats, particularly CA:TG and GA:TC (Clark *et al*, 2011; Kim *et al*, 2011, 2013; Potenski *et al*, 2014). These repeats occur in our retrotransposition reporters at rates similar to the yeast reporters, and as such mutations are likely to inactivate the drug selectable marker used in our assays, this could explain the apparent reduction in retrotransposition in RNase H2 null cells. We therefore set out to determine the occurrence of such mutations in newly inserted LINE-1 elements in RNase H2 null and control cells. To do this, we transfected plasmid JM101/L1.3, containing the *mneoI* retrotransposition cassette, into RNase H2 null HeLa clones (KO1 and KO2) and parental cells, and allowed cells to grow for 5 days without G418 selection (Appendix Fig S3A and B). Two and five days after transfection, genomic DNA was isolated and analysed by conventional PCR, using intron-spanning primers and thus allowing

us to distinguish retrotransposed products (shorter amplification products) from the transfected vector (Appendix Fig S3A and C). Sequencing of amplification products corresponding to the spliced *mneoI* reporter (i.e. *de novo* L1 insertions) showed no increase in mutations in RNASEH2A-KO cells compared to RNase H2 proficient cells (Appendix Fig S3D and E). Notably, only missense mutations were identified, with no 2–5-bp deletions detected in any of the clones analysed. We therefore conclude that the LINE-1 retrotransposition defect in RNase H2 null cells is not caused by hypermutation of *de novo* L1 insertions that could result from failure to remove ribonucleotides misincorporated during TPRT.

### SoF RNase H2 overexpression supports increased LINE-1 retrotransposition, despite reduced substrate affinity

We reasoned that overexpression of the RNase H2 SoF mutant may compensate for its reduced activity against RNA:DNA hybrids and tested the effect of simultaneous overexpression of RNASEH2A-P40D/Y210A, RNASEH2B and RNASEH2C on LINE-1 retrotransposition. We found that overexpression of SoF RNase H2 indeed leads to increased LINE-1 retrotransposition compared to the β-arrestin control (Fig 6A, *P* = 0.019). To further investigate why the separation of function mutant failed to rescue retrotransposition in the complemented RNASEH2A-KO cells, whereas its overexpression did support a higher rate of retrotransposition, we compared enzyme kinetics for SoF and wild-type RNase H2 on RNA:DNA substrate (Fig EV4I). We established that the SoF mutant has much reduced substrate affinity ($K_m^{SoF} \sim 16 \times K_m^{WT}$), whereas its maximum substrate conversion rate is similar to that of WT RNase H2 ($k_{cat}^{SoF} \sim 0.83 \times k_{cat}^{WT}$). The reduced substrate affinity of SoF RNase H2 therefore provides a likely explanation for our observations. These findings are in keeping with a role for the activity of RNase H2 against RNA:DNA hybrids to support LINE-1 retrotransposition.

### RNase H1 overexpression partially rescues LINE-1 retrotransposition in RNase H2 null cells

We assume RNase H1 to be expressed at normal levels in RNASEH2A-KO cells, and because of the marked reduction in LINE-1 retrotransposition in such null cells, it is unlikely that RNase H1

**Figure 6.  Cellular RNase H activity against RNA:DNA hybrids is important for L1 retrotransposition.**

A    Increased L1 retrotransposition upon RNase H2 SoF overexpression. Left, representative retrotransposition and toxicity assays. HeLa cells were transfected with vectors containing an active LINE-1 (WT-hL1, L1.3), alongside an expression vector for β-arrestin as a negative control (−ve), the three RNase H2 subunits (RNase H2, at a 1:1:1 ratio; with RNASEH2A-WT or SoF) or a plasmid expressing APOBEC3A, known to restrict LINE-1 retrotransposition, as a positive control (+ve). Right panel, quantification of this retrotransposition assay, with the level in cells co-transfected with β-arrestin set at 100% for comparison. Values were normalised for transfection efficiency and toxicity. Mean ± SD for *n* = 2 technical replicates (representative of three independent experiments). Unpaired two-sided *t*-test; **P* < 0.05.

B    RNase H activity against RNA:DNA heteroduplexes in RNASEH2A-KO cells (clones KO1 and KO2) is partially rescued by overexpression of human nuclear RNase H1 (KO1 + H1 and KO2 + H1 vs. KO1 + EV and KO2 + EV). RNA:DNA heteroduplex (18 bp; RNA strand 3′-labelled) was incubated with whole-cell lysate and speed of cleavage determined using a FRET-based assay. Mean values ± SEM for *n* = 6 independent experiments.

C    RNase H activity against single-embedded ribonucleotides in RNASEH2A-KO cells (clones KO1 and KO2) is not rescued by human RNase H1 (KO1 + H1 and KO2 + H1 vs. KO1 + EV and KO2 + EV). Speed of cleavage of an 18-bp substrate was determined using a FRET-based assay. Mean values ± SEM for *n* = 3 independent experiments.

D, E    Representative retrotransposition and toxicity assays (D) conducted in the RNASEH2-KO clones (KO1 and KO2) complemented with RNASEH2A-WT (+RNASEH2A) or with human RNase H1 (+RNASEH1). Cells were transfected with vectors containing an active LINE-1 (WT-hL1, L1.3), an RT-mutant LINE-1 (RTm-hL1, D702A), or a toxicity control plasmid (CTRL, pcDNA6.1). Quantification of L1-WT retrotransposition (E). For comparison, the retrotransposition level in KO1 or KO2 cells was set at 100%. Mean ± SD for *n* = 3 technical replicates (representative of three independent experiments). Unpaired two-sided *t*-test; ***P* < 0.01; ****P* < 0.001

F    Proposed model for TPRT. Degradation of LINE-1 RNA in the RNA:cDNA hybrid by cellular RNase H2 allows completion of LINE-1 insertion.

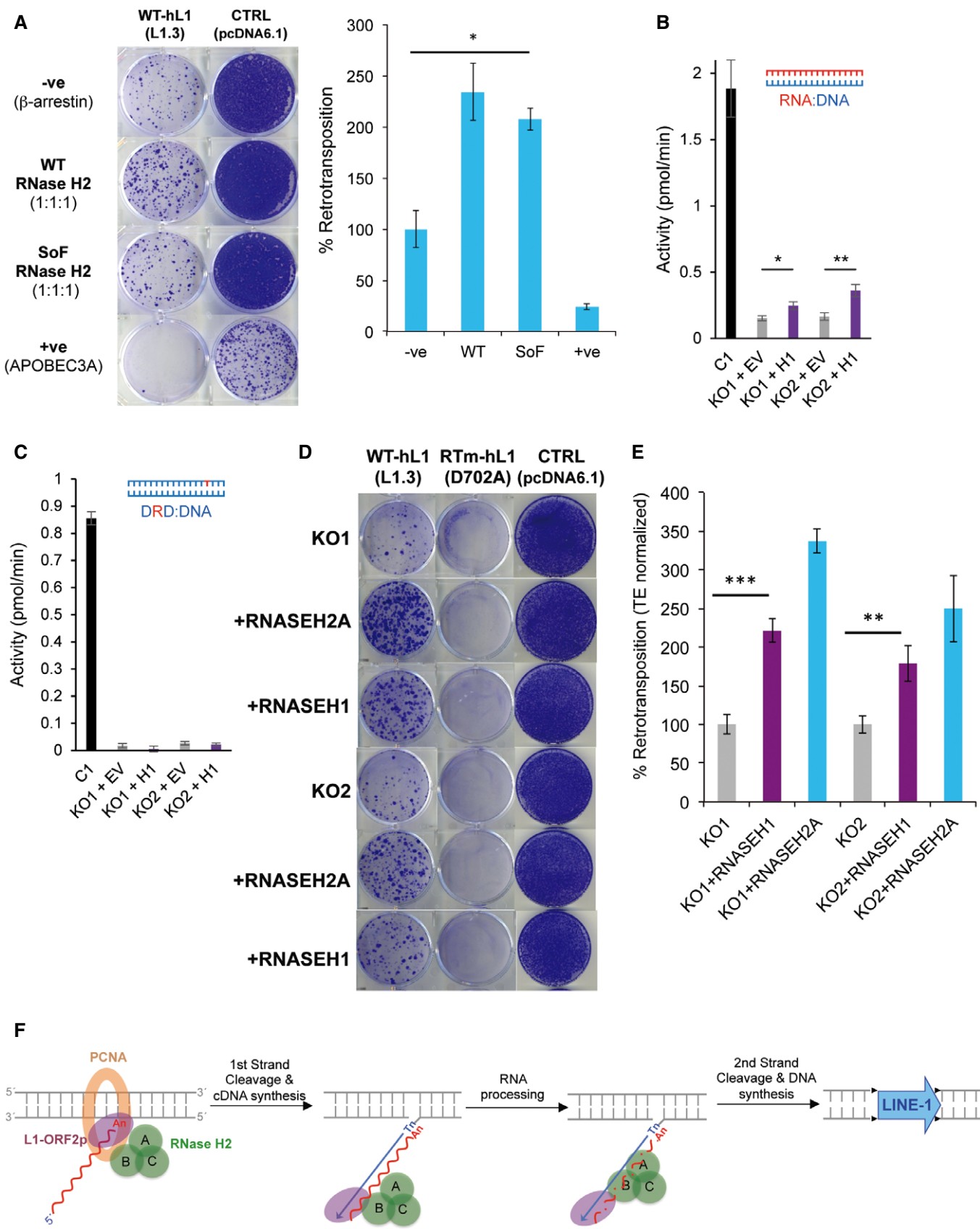

Figure 6.

plays a major role in L1 retrotransposition. However, its activity against RNA:DNA hybrids may still contribute and could explain the variable level of remaining retrotransposition observed in our different RNase H2 null cells; retrotransposition was reduced between 4- and 15-fold compared to parental cells, depending on the cell line used. We therefore tested the effect of complementing HeLa RNASEH2A-KO cells with RNase H1. Two independent HeLa RNASEH2A-KO clones (KO1 and KO2) were transduced with a retroviral vector expressing the nuclear isoform of human RNase H1. To confirm overexpression of RNase H1, activity against RNA:DNA heteroduplexes was measured in lysates of these cells, showing a small but significant increase in enzyme activity (Fig 6B and Appendix Fig S4); RNASEH2A-KO1 cells complemented with wild-type RNase H1 displayed a 1.6-fold increase, while RNASEH2A-KO2 displayed a 2.2-fold increase in cleavage of RNA:DNA heteroduplexes, when compared to the corresponding EV complemented cells. As expected, no effect on RNase H activity against single-embedded ribonucleotides was observed in the RNase H1 complemented cells (Fig 6C). Next, we performed L1 retrotransposition assays using vector JJ101/L1.3, with RNASEH2A-complemented cells as positive controls. Notably, RNase H1 complementation partially alleviated the L1 retrotransposition defect seen in the parental RNASEH2A-KO clones, but not to the same extent as RNASEH2A complementation (3.4-fold increase for RNASEH2A vs. 2.2-fold increase for RNase H1 compared to parental KO1; 2.5-fold vs. 1.8-fold increase, respectively, for KO2, Fig 6D and E). Controls revealed that similar numbers of blasticidin-resistant colonies were formed for all cell lines when transfected with the control plasmid pcDNA6.1 (Fig 6D). Due to the relatively low increase in cellular RNase H activity against RNA:DNA heteroduplexes, failure to fully rescue retrotransposition is perhaps not surprising. However, the ability of RNase H1 to partially rescue L1 activity is consistent with the possibility that both cellular RNase H enzymes could facilitate L1 retrotransposition and would be in line with a model in which these nucleases degrade the RNA:cDNA hybrid formed during TPRT. We therefore propose that RNase H2 facilitates LINE-1 retrotransposition by removing LINE-1 RNA after reverse transcription, allowing completion of the LINE-1 insertion event (Fig 6F), with a minor contribution from nuclear RNase H1.

### Reduced LINE-1 retrotransposition due to RNase H2 disease mutations

Overall, our data suggest that LINE-1 retrotransposition is likely to be reduced in cells from AGS patients with RNase H2 mutations, as most of these are known to cause reduced cellular enzyme activity. We confirmed the impact of disease mutations in any of the three subunits using recombinant RNase H2, but did not find the same change in cleavage pattern observed for the separation of function mutant (Fig EV5A and B). We speculate that the altered cleavage pattern for the latter may be due to its reduced substrate affinity, an effect that may be more pronounced closer to the substrate 3′ end. To determine the cellular effect of RNase H2 disease mutations, we complemented RNASEH2A-KO cells with RNASEH2A-G37S and RNASEH2A-E225G, the only two missense mutations that have been found so far as causative homozygous changes in the catalytic subunit in AGS patients (Rice *et al*, 2013). Western blot analyses revealed that these cells express mutant RNASEH2A (Fig 7A),

leading to a small, but significant increase in cellular RNase H2 activity (Fig 7B). Notably, we observed partial rescue of LINE-1 retrotransposition in cells complemented with RNASEH2A-G37S and RNASEH2A-E225G (Fig 7C and D). Importantly, both AGS mutants displayed significantly reduced retrotransposition compared to RNASEH2A-WT complemented cells ($n = 3$, $P = 0.003$ for G37S, and $P = 0.018$ for E225G). Therefore, based on our data, we would expect AGS patients with RNase H2 mutations to have reduced levels of productive LINE-1 retrotransposition.

## Discussion

RNase H2 has been suggested to control LINE-1 retrotransposition (Volkman & Stetson, 2014), similar to other AGS genes. While our work was under revision, Choi *et al* (2018) published work that suggests that RNase H2 may indeed act as a LINE-1 restriction factor. However, contrary to these findings, our results, using multiple independent CRISPR/Cas9 edited RNase H2 null clones, three different cell lines and several engineered LINE retrotransposition reporters, provide comprehensive evidence to support a role for cellular RNase H2 activity in promoting LINE-1 retrotransposition. Furthermore, our findings are consistent with a recent report that also found RNase H2 to be required for retrotransposition (Bartsch *et al*, 2017). We also demonstrate that other retrotransposons lacking an RNase H domain also rely on cellular RNase H activity to mobilise. In contrast, retrotransposons that code for an RNase H domain/activity did not require cellular RNase H2 to retrotranspose efficiently. Complementation of the retrotransposition defect in RNase H2 null cells by RNase H1 overexpression and increased retrotransposition upon overexpression of the RNase H2 separation of function mutant lead us to propose a model in which RNase H2 degrades LINE-1 RNA in the RNA:cDNA retrotransposition intermediate generated during TPRT (Fig 6F).

Recently, a PCNA Interaction Protein (PIP) motif was identified in L1-ORF2p (Taylor *et al*, 2013), and the interaction between L1-ORF2p and PCNA was shown to be required for efficient retrotransposition. As RNASEH2B also contains a functional PIP domain that allows RNase H2 to interact with PCNA (Chon *et al*, 2009; Bubeck *et al*, 2011), we speculate that PCNA might act as an anchor protein connecting L1-ORF2p with RNase H2 during retrotransposition. This may also explain why RNase H1, which lacks a PIP domain, cannot efficiently complement the L1-retrotransposition defect inherent to RNase H2 null cells. However, nuclear RNase H1 could process LINE-1 RNA:cDNA hybrids by a simple but less efficient diffusion mechanism. Whether the interaction between PCNA and RNase H2 indeed contributes to efficient retrotransposition remains to be determined.

Our model, in which cellular RNase H activity promotes completion of LINE-1 retrotransposition by degrading the RNA from RNA:cDNA hybrids generated during TPRT (Fig 6F), would explain how LINE-1 elements can function without an active RNase H domain (Malik *et al*, 1999; Olivares *et al*, 2002). We provided several lines of evidence supporting this model: (i) strongly reduced LINE-1 retrotransposition in RNase H2 null cells; (ii) rescue of this defect by wild-type RNASEH2A as well as (iii) RNase H1 overexpression; and (iv) increased retrotransposition upon overexpression of both wild-type and separation of function RNase H2. Our work shows that

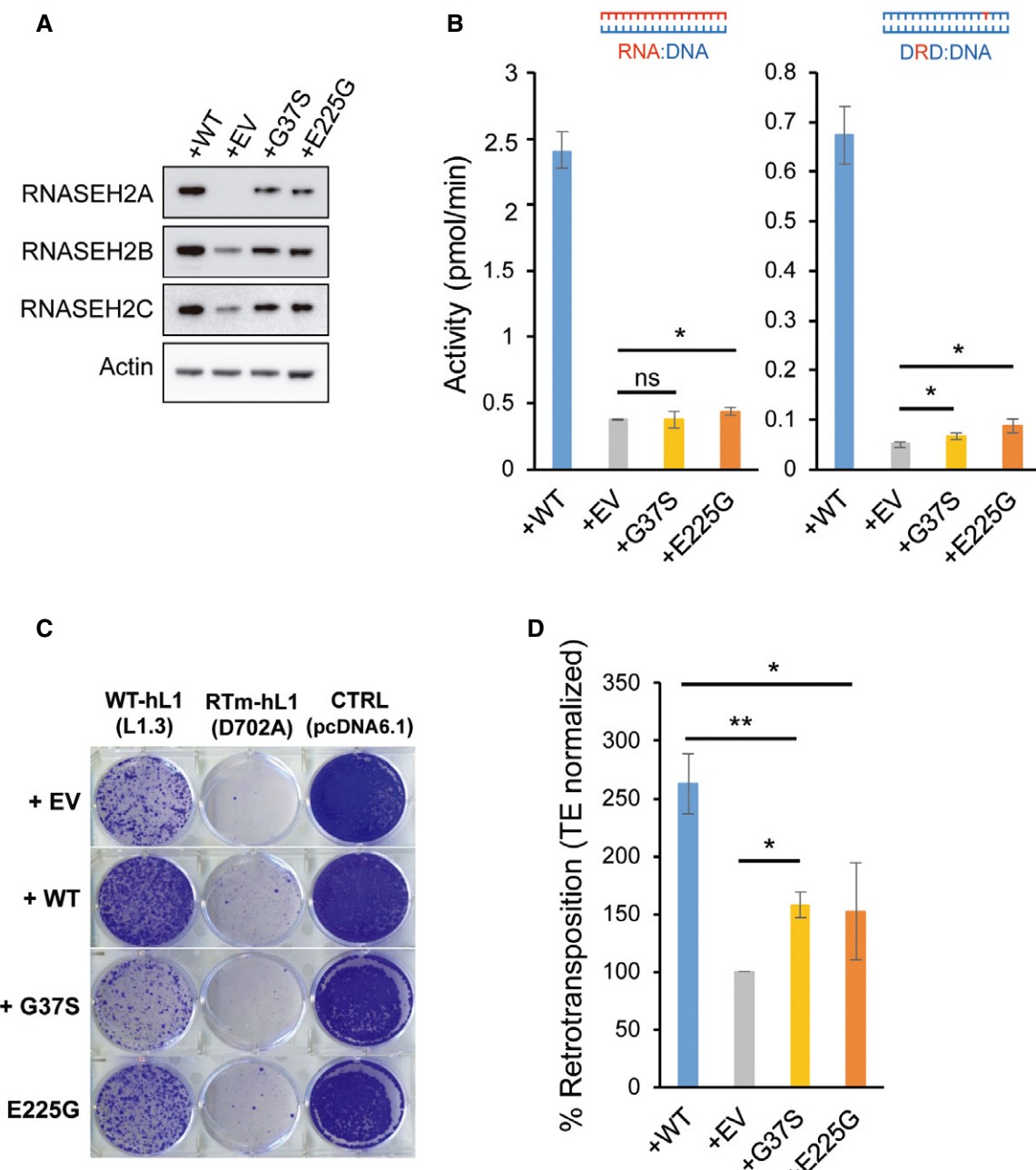

**Figure 7. Reduced L1 retrotransposition due to RNase H2 AGS mutations.**

A    Western blot analysis of RNase H2 expression in RNASEH2A-KO HeLa cells (KO2) complemented with the indicated retroviral vector (EV, empty vector; WT, wild-type RNASEH2A; RNASEH2A-G37S; RNASEH2A-E225G). Actin was used as a loading control.

B    Complementation of RNASEH2A-KO cells with RNASEH2A with AGS mutations (G37S and E225G) leads to a small but significant increase in RNase H activity. Mean values $\pm$ SEM for $n = 3$ independent experiments. Unpaired two-sided $t$-test; $*P < 0.05$; ns, $P > 0.05$.

C, D  Cells expressing AGS mutant RNase H2 fail to support efficient L1 retrotransposition. (C) Representative retrotransposition and toxicity assays conducted in the four complemented lines. Cells were transfected with vectors containing an active human LINE-1 (WT-hL1, L1.3), an RT-mutant LINE-1 (RTm-hL1, D702A) or a toxicity control plasmid (CTRL, pcDNA6.1). (D) Quantification of L1-WT retrotransposition in the complemented lines. For comparison, the retrotransposition level in KO cells complemented with empty vector (EV) was set at 100%. Mean $\pm$ SD for $n = 3$ independent experiments (each experiment performed in technical duplicates). Unpaired two-sided $t$-test; $*P < 0.05$; $**P < 0.01$.

Source data are available online for this figure.

RNase H activity directed against RNA:DNA hybrids, mainly provided by cellular RNase H2, is important for efficient and productive LINE-1 retrotransposition. We interpret this to mean that it is involved in degrading LINE-1 RNA in the RNA:cDNA hybrid, allowing second-strand synthesis and ultimately insertion into the genome. Alternative explanations are of course possible. As there is

increasing evidence that DNA double-strand break repair can be mediated by RNA (Keskin *et al*, 2014; Ohle *et al*, 2016; Michelini *et al*, 2017), one intriguing possibility is that RNA:DNA hybrids play a more active role in the LINE-1 retrotransposition process, for example by recruiting RNase H2 and DNA repair machinery. Furthermore, RNase H2 deficiency can also cause larger genomic rearrangements (Reijns *et al*, 2012) which might impact on LINE-1 retrotransposition. Although we cannot rule out that this type of genome instability interferes with retrotransposition, or even that non-productive retrotransposition contributes to increased genomic rearrangements in RNase H2 null cells, it seems unlikely that this is the reason for the reduced LINE-1 retrotransposition we observe, particularly as RNase H2 overexpression leads to increased rates of retrotransposition.

As mutations in the genes encoding RNase H2 are a frequent cause of AGS, our findings are relevant with regard to possible sources of immunostimulatory nucleic acids thought to cause autoinflammation. Two sources for such cytoplasmic nucleic acids have been proposed: DNA damage or retroelements. Notably, active LINE-1s are expressed and highly active in the central nervous system (Muotri *et al*, 2005; Coufal *et al*, 2009) and strong experimental evidence suggests that TREX1, SAMHD1 and ADAR1 act as LINE-1 restriction factors (Stetson *et al*, 2008; Zhao *et al*, 2013; Orecchini *et al*, 2017; Thomas *et al*, 2017). Although one study failed to detect elevated retrotransposition of L1s in the hippocampus of an AGS patient with SAMHD1 mutations (Upton *et al*, 2015), more recent work using TREX1-deficient neural cells generated from human embryonic stem cells has strongly implicated accumulating LINE-1-derived single-stranded DNAs (ssDNAs) in type I IFN production and neurotoxicity (Thomas *et al*, 2017), consistent with a role for active LINE-1s in AGS pathophysiology (Garcia Perez & Alarcon-Riquelme, 2017). This work also reinforces the concept that by-products of active retrotransposition, rather than the accumulation of LINE-1 insertions per se, may be the driver of AGS pathology (Upton *et al*, 2015), with TREX1 normally degrading LINE-1 ssDNA retrotransposition intermediates (Stetson *et al*, 2008; Garcia Perez & Alarcon-Riquelme, 2017; Thomas *et al*, 2017).

Very recently, we have demonstrated both increased DNA damage and cGAS-STING dependent upregulation of interferon-stimulated genes (ISG) in RNase H2 null MEFs (Mackenzie *et al*, 2016). Furthermore, we established a mechanism linking genome instability to inflammation, with micronuclei providing a source of cytoplasmic DNA able to activate cGAS, (Mackenzie *et al*, 2017), although the direct relevance of this to AGS remains to be determined. Notably, neither DNA damage nor the ISG response was alleviated by overexpression of RNase H1 (Mackenzie *et al*, 2016), in contrast to partial rescue of the retrotransposition defect in RNase H2 null HeLa cells overexpressing RNase H1 (Fig 6). We therefore favour the possibility that genome instability is the underlying cause of autoinflammation in AGS associated with RNase H2 mutations. However, because RNase H2 activity is required for LINE-1 retrotransposition and we show that cells with RNase H2 disease mutations have reduced levels of productive retrotransposition, it is formally possible that accumulation of nuclear LINE-1 RNA:cDNA hybrids is a source of immunostimulatory nucleic acids. How such hybrids that are covalently linked to the genome would access the cytoplasm and activate pattern recognition receptors is currently unclear though. An additional remaining question is whether

physiological levels of retrotransposition by-products are sufficient to elicit the observed inflammatory response. Clearly, further work is therefore needed to determine the relative importance of retroelement activity and genome instability in AGS. This will not be straight forward, as the two are not necessarily mutually exclusive.

In summary, our work contributes to the mechanistic understanding of LINE-1 retrotransposition, as we demonstrate that cellular RNase H2 plays an integral part in LINE retrotransposition, explaining how LINE elements lacking an RNase H domain can retrotranspose. In addition, our data add a new layer of complexity to the understanding of AGS pathophysiology, as we demonstrate that not all AGS proteins are LINE restriction factors.

# Materials and Methods

### Cell culture

HeLa cells, a kind gift from G. Stewart (Birmingham) originally obtained from ATCC, were grown in Dulbecco's modified Eagle's medium (DMEM; Gibco, Cat no 41965-039) supplemented with 10% foetal bovine serum (FBS), 50 U/ml penicillin and 50 μg/ml streptomycin. U2OS cells purchased from the European Collection of Authenticated Cell Cultures (ECACC, Cat no. 92022711) and HCT116 p53$^{-/-}$ cells (Bunz *et al*, 1998; Dornan *et al*, 2004), a kind gift from K. Ball (Edinburgh), were maintained in modified McCoy's 5A medium (Gibco; Cat no 26600-023) supplemented with 10% FBS, 50 U/ml penicillin and 50 μg/ml streptomycin.

All cell lines were grown at 37°C, 5% $CO_2$ and atmospheric $O_2$ and passaged using Trypsin (Gibco). Checks were performed at least once a month using the Lonza-Mycoalert Mycoplasma Detection Kit to ensure that all cells were mycoplasma-free. In addition, the identity of the cell lines was confirmed by SRT analyses at least once a year (Lorgen, Granada, Spain).

### Plasmid DNA

All plasmids were purified using a Plasmid Midi kit from Qiagen. DNA was analysed by electrophoresis (0.7% agarose-ethidium bromide gels), and only highly supercoiled DNA preparations were used in transfection experiments. Cloning strategies are available upon request.

*pSpCas9n(RNASEH2A-g1)-2A-GFP* and *pSpCas9n(RNASEH2A-g2)-2A-Puro* express guide RNAs (gRNAs) designed against exon 1 (TGCCCGCCTCATCGACGCCC) and intron 1 (CCCGTGCTGGGTGCGCCCCT) of human RNASEH2A, and Cas9n (D10A nickase mutant) fused to the cDNA of Enhanced Green Fluorescent Protein (EGFP) and puromycin N-acetyl-transferase (puro), respectively. gRNAs were generated by annealing DNA oligonucleotides and were cloned into the BbsI site of pSpCas9n(BB)-2A-GFP and pSpCas9n(BB)-2A-Puro vectors (Addgene plasmids #48140 and #48141, respectively; gifts from Feng Zhang) as previously described (Ran *et al*, 2013).

*pMSCVpuro-RNASEH2A-WT*, coding sequence of human RNASEH2A (NM_006397.2) cloned into vector pMSCVpuro-Dest, a Gateway compatible version of pMSCVpuro (Clontech).

*pMSCVpuro-RNASEH2A-SoF*, a derivative of plasmid pMSCVpuro-RNASEH2A-WT that contains two missense mutations

in the coding sequence of human RNASEH2A (P40D/Y210A, separation of function mutant, Chon *et al*, 2013).

*pMSCVpuro-RNASEH2A-CD*, a derivative of plasmid pMSCVpuro-RNASEH2A-WT that contains two missense mutations in the coding sequence of human RNASEH2A (D34A/D169A, catalytically inactive, Reijns *et al*, 2011).

*pMSCVpuro-RNaseH1*, coding sequence of the nuclear isoform of the human RNASEH1 gene (NM_002936.5, aa27–286) cloned into pMSCVpuro-Dest.

*pcDNA3.1/nV5-RNASEH2A*, coding sequence of human RNASEH2A cloned into vector pcDNA3.1/nV5-DEST (contains an N-terminal V5 tag).

*pcDNA3.1/nV5-RNASEH2A-SoF*, coding sequence of human RNASEH2A with P40D/Y210A missense mutations cloned in vector pcDNA3.1/nV5-DEST (contains an N-terminal V5 tag).

*pcDNA3.1/nV5-RNASEH2B*, coding sequence of human RNASEH2B (NM_024570.3) cloned into pcDNA3.1/nV5-DEST (contains an N-terminal V5 tag).

*pcDNA3.1/nV5-RNASEH2C*, contains the coding sequence of the human RNASEH2C gene (NM_032193.3) cloned in pcDNA3.1/nV5-DEST (contains an N-terminal V5 tag).

*pK-βarr*, described previously (Bogerd *et al*, 2006) and expresses C-terminally HA-tagged human β-arrestin.

*pK-A3A*, described previously (Bogerd *et al*, 2006) and expresses C-terminally HA-tagged human APOBEC3A.

*pU6ineo*, described previously (Richardson *et al*, 2014). Contains the neomycin phosphotransferase (NEO) expression cassette from pEGFP-N1 (Clontech) cloned into a modified pBSKS-II(+) (Stratagene) that contains a U6 promoter in the multi-cloning site.

*pcDNA6.1*, (Invitrogen) contains an expression cassette for blasticidin S deaminase.

*JM101/L1.3*, described previously (Sassaman *et al*, 1997) and contains a full-length copy of the human L1.3 element tagged with the *mneoI* indicator cassette (Freeman *et al*, 1994; Moran *et al*, 1996) and is cloned in pCEP4 (Life Technologies).

*JM101/L1.3-D205A*, described previously (Wei *et al*, 2001); a derivative of JM101/L1.3 that contains a missense mutation in the EN domain of L1-ORF2p (D205A).

*JM101/L1.3-D702A*, was described previously (Wei *et al*, 2001); a derivative of JM101/L1.3 that contains a missense mutation in the RT domain of L1-ORF2p (D702A).

*JJ101/L1.3*, described previously (Kopera *et al*, 2011). It contains a full-length copy of the human L1.3 element (Sassaman *et al*, 1997) tagged with the *mblastI* indicator cassette (Morrish *et al*, 2002; Goodier *et al*, 2007) and is cloned in pCEP4 (Life Technologies).

*JJ101/L1.3-D205A*, described previously (Kopera *et al*, 2011); a derivative of JJ101/L1.3 that contains a missense mutation in the EN domain of L1-ORF2p (D205A).

*JJ101/L1.3-D702A*, described previously (Kopera *et al*, 2011); a derivative of JJ101/L1.3 that contains a missense mutation in the RT domain of L1-ORF2p (D702A).

*pXY014*, described previously (Xie *et al*, 2011); contains a full-length copy of the human L1RP element (Kimberland *et al*, 1999) tagged with the *mflucI* indicator cassette (Xie *et al*, 2011) and is cloned in a modified pCEP4 (Life Technologies) that contains a Renilla firefly expression cassette.

*pXY017*, described previously (Xie *et al*, 2011); a derivative of pXY014 that contains two missense mutations in the RNA

binding domain of L1-ORF1p (RR261/62AA). This plasmid was used as a negative control of the luciferase-based retrotransposition assays.

*pCMVMusD-6neo^TNF*, described previously (Ribet *et al*, 2004); contains a full-length copy of a mouse MusD element (AC124426, positions 9,078–16,569 (+)) tagged with the neo$^{TNF}$ indicator cassette (Esnault *et al*, 2002) and is cloned in vector pCMVbeta (Clontech).

*Zfl2-2mneoI*, described previously (Sugano *et al*, 2006; Garcia-Perez *et al*, 2010); contains a full-length copy of the zebrafish Zfl2-2 element tagged with the *mneoI* indicator cassette inside the 3′UTR of the LINE (Freeman *et al*, 1994; Sugano *et al*, 2006) and is cloned in pCEP4 (Life Technologies).

*pT2neo*, described previously (Mates *et al*, 2009); contains an SV40-driven neomycin phosphotransferase cDNA flanked by SB TIRs.

*pCMV-SB100x*, described previously (Mates *et al*, 2009); contains a CMV-driven hyperactive SB Transposase.

*pCEP-EGFP*, described previously (Alisch *et al*, 2006); contains the coding sequence of the humanised GFP protein cloned in pCEP4 (Invitrogen).

*pGEX6P1-hsRNASEH2BCA*, *pGEX6P1-hsRNASEH2BCA(D34A/ D169A)*, *pGEX6P1-hsRNASEH2B(A177T)CA* and *pGEX6P1-hsRNA-SEH2BC(R69W)A*, described previously (Reijns *et al*, 2011) allow expression in *Escherichia coli* of GST-tagged human RNASEH2B and non-tagged RNASEH2C and A subunits. Amino acid substitutions indicated in brackets were introduced into the relevant subunits by site-directed mutagenesis.

*pGEX6P1-hsRNASEH2BCA(P40D/Y210A)*, *pGEX6P1-hsRNA-SEH2BCA(G37S)* and *pGEX6P1-hsRNASEH2BCA(E225G)* had the P40D/Y210A separation of function mutations, and G37S and E225G AGS mutations respectively introduced into RNASEH2A by site-directed mutagenesis.

## Generation of RNASEH2A knockout cell lines

To establish RNASEH2A-KO cell lines, cells were seeded in 6-well plates and transfected with the two vectors encoding both the sgRNAs and Cas9n using Lipofectamine 2000 (Thermo Fisher Scientific). Forty-eight hours after transfection, single EGFP-expressing cells were sorted into 96-well plates on a BD FACSJazz instrument (BD Biosciences) and grown until cell lines formed. RNASEH2A-KO clones were selected on the basis of the size of PCR products of the targeted region, and deletions/insertions subsequently confirmed by Sanger DNA sequencing. Oligonucleotides (5′–3′) used for PCR amplification and sequencing of targeted RNASEH2A loci were 5′-ACCCGCTCCTGCAGTATTAG and 5′-TCCCTTGGTGCAGTGCAATC. The absence of functional RNASEH2A was confirmed by immunoblotting, by an RNase H2 activity assay, and using alkaline gel electrophoresis as described below. Only clones confirmed to be functionally null were used for subsequent experiments. Some knockout clones retained very low levels of RNASEH2A protein expression, apparent upon long exposure of immunoblots. For these clones, Sanger sequencing showed the presence of in-frame deletions, in each case removing essential catalytic site residues, including Asp34, rendering them enzymatically non-functional. Clones expressing wild-type RNASEH2A protein were identified in parallel and used as controls.

## Retroviral complementation

To complement RNASEH2A-KO HeLa clones, cells were infected with retroviral supernatant produced in Amphotropic Phoenix packaging cells (Swift *et al*, 2001) using pMSCVpuro-based vectors, in the presence of 4 μg/ml polybrene and selected for stable integration using 2 μg/ml puromycin.

## Whole-cell extracts preparation and Western blot analysis

Whole-cell extracts (WCE) for RNase H activity assays and for determining protein levels of RNase H2 subunits were prepared by incubating cells in lysis buffer [50 mM Tris–HCl pH 8.0, 280 mM NaCl, 0.5% NP-40, 0.2 mM EDTA, 0.2 mM EGTA, 10% glycerol (vol/vol), 1 mM DTT and 1 mM phenylmethyl-sulfonyl fluoride (PMSF)] for 10 min on ice, followed by the addition of an equal volume of 20 mM HEPES pH 7.9, 10 mM KCl, 1 mM EDTA, 10% glycerol (vol/vol), 1 mM DTT and 1 mM PMSF for an additional 10 min. Whole-cell extracts were cleared by centrifugation (17,000 *g* for 10 min at 4°C), and protein concentration was determined by Bradford assay (Protein Assay Kit, Bio-Rad).

In overexpression assays conducted with HeLa and U2OS cells, WCEs were prepared using RIPA buffer (Sigma) supplemented with 1× Complete Mini EDTA-free Protease Inhibitor cocktail (Roche), 0.1% Phosphatase Inhibitor 1&2 (Sigma), 1 mM (PMSF) (Sigma) and 0.25% β-mercaptoethanol (Sigma), incubating cells for 10 min on ice. Cellular debris was removed by centrifugation (1,000 *g* for 5 min at 4°C) and total protein concentration was determined using the Micro BCA Protein Assay Kit (Thermo) following standard procedures.

Equal amounts of protein lysates were run on SDS–PAGE and transferred to PVDF or nitrocellulose membranes (Bio-Rad). Membranes were blocked in 5% milk/TBST [TBS + 0.2% Tween-20 (v/v)] and incubated with primary antibodies diluted in 5% milk/TBST overnight at 4°C. Membranes were then washed 3 times with TBST, incubated with secondary antibodies for 1 h at RT, washed again and developed. As secondary antibodies, we used horseradish peroxidase (HRP)-linked antibodies (Cell Signaling) and either Amersham ECL Prime Western Blotting Detection Reagent (GE Healthcare Life Sciences) or an Inmun-StarTM Western CTM Detection Kit (BIO-RAD). The light signal was captured on X-ray films or using an ImageQuantLAS4000 device following manufacturer's recommendations. To quantify L1-ORF1p expression levels, we used an infrared fluorescent detection system (Odyssey, LI-COR) following manufacturer's recommendations (Macia *et al*, 2017).

The following antibodies were used for immunoblotting (at indicated dilutions): sheep anti-RNase H2 (raised against human recombinant RNase H2, Reijns *et al*, 2012, 1:1,000); rabbit anti-RNASEH2A (Origene TA306706, 1:1,000) or mouse anti-RNASEH2A (Santa Cruz sc-515475, 1:1,000); mouse anti-α-tubulin B512 (Sigma T6074, 1:5,000); mouse anti-vinculin (Sigma V9264, 1:1,000); rabbit anti-L1Hs-ORF1p (provided by Dr. Oliver Weichenrieder, Max-Planck, Germany, Macia *et al*, 2017, 1:5,000); mouse anti-β-actin (1:20,000; Sigma); mouse anti-V5 (clone V5-10, Sigma V8012, 1:10,000). In quantitative westerns, goat anti-rabbit and anti-mouse fluorescent secondary antibodies were used at a 1:20,000 dilution.

## RNase H2 activity assay

To assess RNase H2 activity in whole-cell extracts, a FRET-based fluorescent substrate release assay was performed as previously described (Reijns *et al*, 2011). Briefly, RNase H2-specific activity was determined by measuring cleavage of a single-embedded ribonucleotide-containing double-stranded DNA substrate (DRD: DNA). Activity against a DNA:DNA substrate of the same sequence was used to correct for non-RNase H2 "background activity" against the DRD:DNA substrate. Substrates were formed by annealing a 3′-fluorescein-labelled oligonucleotide (5′GATCTGAGCCTGGGaGCT or 5′GATCTGAGCCTGGGAGCT; uppercase DNA, lowercase RNA) to a complementary 5′ DABCYL-labelled DNA oligonucleotide (Eurogentec). Reactions were performed in 100 μl of reaction buffer (60 mM KCl, 50 mM Tris–HCl pH 8.0, 10 mM $MgCl_2$, 0.01% BSA, 0.01% Triton X-100) with 250 nM substrate in 96-well flat-bottomed plates at 24°C. Whole-cell lysates were prepared as described above, and the final protein concentration used per reaction was 100 ng/μl. Fluorescence was read (100 ms) every 5 min for up to 90 min using a VICTOR2 1420 multilabel counter (Perkin Elmer), with a 480-nm excitation filter and a 535-nm emission filter.

To assess RNase H2 activity in whole-cell extracts using the gel-based assay, a range of protein concentrations (50–400 ng/μl) was incubated with 2 μM substrate (described above; 5′gatctgagc-ctgggagct for RNA:DNA) in 5 μl reactions at 37°C for 30 min or 1 h. Reactions were stopped through addition of an equal volume of 96% formamide, 20 mM EDTA and heating at 95°C. Products were resolved by denaturing PAGE (20%, 1× TBE), visualised on a FLA-5100 imaging system (Fujifilm) and quantified using ImageQuant TL (GE Healthcare).

## Detection of ribonucleotides in genomic DNA

Total nucleic acids were isolated from ~1 million cells by lysis in ice-cold buffer (20 mM Tris–HCl pH 7.5, 75 mM NaCl, 50 mM EDTA) and subsequent incubation with 200 μg/ml proteinase K (Roche) for 10 min on ice followed by addition of N-lauroylsarcosine sodium salt (Sigma) to a final concentration of 1%. Nucleic acids were sequentially extracted with TE-equilibrated phenol, phenol:chloroform:isoamyl alcohol (25:24:1) and chloroform, then precipitated with isopropanol, washed with 75% ethanol and dissolved in nuclease-free water.

For alkaline gel electrophoresis, 500 ng of total nucleic acids was incubated with 1 pmol of purified recombinant human RNase H2 (Reijns *et al*, 2011) and 0.25 μg of DNase-free RNase (Roche) for 30 min at 37°C in 100 μl reaction buffer (60 mM KCl, 50 mM Tris–HCl pH 8.0, 10 mM $MgCl_2$, 0.01% BSA, 0.01% Triton X-100). Nucleic acids were ethanol precipitated, dissolved in nuclease-free water and separated on 0.7% agarose in 50 mM NaOH, 1 mM EDTA. After electrophoresis, the gel was neutralised in 0.7 M Tris–HCl pH 8.0, 1.5 M NaCl and stained with SYBR Gold (Invitrogen). Imaging was performed on a FLA-5100 imaging system (Fujifilm), and densitometry plots were generated using an AIDA Image Analyzer (Raytest).

## Retrotransposition assays

LINE-1 retrotransposition assays in RNASEH2A-KO and control clones carried out in HeLa (*n* = 12) and HCT116 p53⁻/⁻ (*n* = 6)

cells were done blindly; in addition, at least 3 independent experiments were conducted per assay. Retrotransposition assays were carried out as previously described (Wei *et al*, 2000; Heras *et al*, 2013). Briefly, cells were plated in 6-well dishes (for retrotransposition and toxicity assays, Corning); when indicated, cells were plated in 10-cm plates (for toxicity assays, Corning). For assays employing human LINE-1-based constructs (plasmid JM101/L1.3 and mutants, JJ101/L1.3 and mutants, and pXY014 and mutants), approximately $2 \times 10^4$ cells were plated per well in a 6-well dish; when employing zebrafish LINEs (plasmid Zfl2-2*mneoI*) and LTR-retrotransposons (plasmid pCMVMusD-6neo$^{TNF}$), approximately $4 \times 10^4$ cells were plated per well in a 6-well dish; in toxicity assays (plasmids pU6ineo or pcDNA6.1), approximately $1 \times 10^4$ cells were plated per well in a 6-well dish; in some toxicity assays, approximately $4 \times 10^4$ cells were plated in 10-cm plates. Eighteen hours after plating, DNA transfections were carried out using FuGene 6 transfection reagent (Promega) and Opti-MEM (Life Technologies) following the protocol provided by the manufacturer (for a six-well plate: 3 μl of FuGene and 97 μl of Opti-MEM and 1 μg of DNA transfected; for a 10-cm plate: 12 μl of FuGene and 388 μl of Opti-MEM and 4 μg of DNA transfected). The day after transfection, media was replaced with fresh media. Luciferase-based retrotransposition assays were analysed 96 h post-transfection using the Dual-Glo® luciferase assay system (Promega) following the protocol provided by the manufacturer. When G418 selection was applied (*mneoI* containing plasmids), cells were subjected to selection with 400 μg/ml G418 (Life Technologies) starting approximately 72 h post-transfection; when blasticidin S selection was applied (*mblastI* containing plasmids), cells were subjected to selection with 5 μg/ml blasticidin S (Life Technologies) starting approximately 120 h post-transfection; G418 selection was conducted for 12 days, while blasticidin S selection was carried out for 7 days. After selection, antibiotic-resistant colonies were fixed with 2% paraformaldehyde/0.4% glutaraldehyde and stained with 0.1% crystal violet solution as described (Moran *et al*, 1996; Wei *et al*, 2000). In all experiments, a co-transfection with plasmid pCEP-EGFP was carried out in parallel with control for transfection efficiency (TE); co-transfected cells were harvested 72 h post-transfection and we used FACS to determine the percentage of EGFP-expressing cells. In the overexpression experiments, to control for toxicity and colony-forming capacity, cells were transfected in parallel with the indicated toxicity vector (either pU6i-NEO or pcDNA6.1) and colony numbers used for additional normalisation as described (Kopera *et al*, 2016). Selection, fixation and staining were conducted as described above.

### Transposition assays

To assay SB transposition in HeLa and U2OS parental cells, and derived RNASEH2A-KO and control clones, approximately $1 \times 10^5$ cells were plated per well in a 6-well dish. Eighteen hours after plating, DNA transfections were carried out using FuGene 6 and Opti-MEM following the protocol provided by the manufacturer (see above). Cells were transfected with 1 μg of plasmid pT2neo or co-transfected with 1 μg of plasmid pT2neo and 0.5 μg of plasmid pCMV-SB100x per well. Seventy-two hours after transfection, cells were trypsinised and counted; next, 10% of the transfected cells were plated on a 10-cm plate and G418 selection started 48 h later (using 400 μg/ml). G418 selection was conducted for 10 days, and

G418-resistant colonies were fixed with 2% paraformaldehyde/0.4% glutaraldehyde and stained with 0.1% crystal violet solution. As in retrotransposition experiments, a co-transfection with plasmid pCEP-EGFP was carried out in parallel to control for TE differences.

### PCR-based mutation analyses

To test whether RNase H2 deficiency caused increased mutation rates in *de novo* inserted L1-sequences, we used a previously described assay (Bogerd *et al*, 2006). Briefly, parental HeLa, HeLa RNASEH2A-KO1 and KO2 cells were plated ($1 \times 10^5$ cells per well in a 6-well dish) and 18 h later transfected with 1 μg of plasmid JM101/L1.3 or with plasmid JM101/L1.3-D702A as a control, using the conditions described above. Forty-eight (2 days) and one hundred and twenty hours (5 days) after transfection, genomic DNA (gDNA) was isolated from transfected cells, digested with *Swa*I (NEB; there is a single *Swa*I site within the intron of the *mneoI* retrotransposition indicator cassette) and used as template in PCRs using primers NEO437s (5′GAGCCCCTGATGCTCTTCGTCC) and NEO1808as (5′CATTGAACAAGATGGATTGCACGC) that flank the engineered intron in *mneoI*. PCRs were carried in 25 μl using KAPA Taq ReadyMIx PCR and 0.4 μM of each primer. DNA-free water (Gibco) was included as a negative control in all assays, as well as PCRs conducted on gDNA isolated from naïve HeLa, and gDNA from naïve HeLa digested with *Swa*I. PCR conditions for NEO amplification were as follows: 1× (95°C, 3 min); 40× (15 s, 95°C; 15 s, 60°C; 30 s, 72°C); 1× (72°C, 1 min). PCR products were resolved on 1.5% agarose gels, amplified products excised, purified and cloned in pGEMT-Easy (Promega). Clones were sequenced using M13FWD primer.

### Statistics

Unless otherwise stated, all statistics were performed using two-sided unpaired *t*-test (parametric) or Mann–Whitney test (non-parametric) comparing replicates as indicated. $P < 0.05$ was considered as statistically significant.

**Expanded View** for this article is available online.

### Acknowledgements

We acknowledge Dr. Marcin Nowotny (IIMCB, Warsaw, Poland) and current members of the J.L.G.-P. laboratory for helpful discussions and critical reading of the manuscript. We also acknowledge Drs. John Goodier (John Hopkins, US) for providing RNASEH2 overexpression plasmids constructs, Oliver Weichenrieder (Max-Planck, Tubingen, Germany) for providing a polyclonal L1-ORF1p antibody, Thierry Heidmann (Institut Pasteur, France) for providing the MusD construct and Zoltan Ivics (PEI, Germany) for providing the Sleeping Beauty vectors used in this study. M.B.-G. is funded by a "Formacion Profesorado Universitario" (FPU) PhD fellowship from the Government of Spain (MINECO, Ref FPU15/03294), and this paper is part of her thesis project ("Epigenetic control of the mobility of a human retrotransposon"). R.V.-A. is funded by a PFIS Fellowship from the Government of Spain (ISCiii, FI16/00413). O.M. is funded by an EMBO Long-Term Fellowship (ALTF 7-2015), the European Commission FP7 (Marie Curie Actions, LTFCOFUND2013, GA-2013-609409) and the Swiss National Science Foundation (P2ZHP3_158709). S.R.H. is funded by the Government of Spain (MINECO, RYC-2016-21395 and SAF2015-71589-P). A.P.J's laboratory is supported by the UK Medical Research Council (MRC

University Unit grant U127527202). J.L.G.P's laboratory is supported by CICE-FEDER-P12-CTS-2256, Plan Nacional de I+D+I 2008-2011 and 2013-2016 (FIS-FEDER-PI14/02152), PCIN-2014-115-ERA-NET NEURON II, the European Research Council (ERC-Consolidator ERC-STG-2012-233764), by an International Early Career Scientist grant from the Howard Hughes Medical Institute (IECS-55007420), by The Wellcome Trust-University of Edinburgh Institutional Strategic Support Fund (ISFF2) and by a private donation from Ms Francisca Serrano (Trading y Bolsa para Torpes, Granada, Spain).

## Author contributions

MB-G and CL-R performed all retrotransposon-related experiments, with the help/advice of SRH, MG-C, RV-A, DC and LS. MAMR performed all RNase H biochemistry experiments. TCW conducted the tandem repeat sequence analyses. All KO and complemented cell lines were generated and characterised by ZT, OM, MMM, AF, M-JHCK and MAMR; and SRH and AS-P provided valuable input throughout the project. MAMR, APJ and JLG-P directed and designed the study. JLG-P and MAMR wrote the manuscript, with input from all authors.

## Conflict of interest

The authors declare that they have no conflict of interest.

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
