## [Review Process File · The EMBO Journal]

RNase H2, mutated in Aicardi-Goutières syndrome, promotes LINE-1 retrotransposition

Maria Benitez-Guijarro, Cesar Lopez-Ruiz, Žygimantė Tarnauskaitė, Olga Murina, Mahwish Mian Mohammad, Thomas C. Williams, Adeline Fluteau, Laura Sanchez, Raquel Vilar-Astasio, Marta Garcia-Canadas, David Cano, Marie-Jeanne H. C. Kempen, Antonio Sanchez-Pozo, Sara R. Heras, Andrew P. Jackson, Martin A. M. Reijns and Jose L. Garcia-Perez

Review timeline:

Submission date:	26 th October 2017
Editorial Decision:	18 th December 2017
Revision received:	20 th April 2018
Editorial Decision:	9 th May 2018
Revision received:	23 rd May 2018
Accepted:	28 th May 2018

Editor: Karin Dumstrei.

Transaction Report:

1st Editorial Decision

18th December 2017

Thank you for submitting your manuscript to The EMBO Journal. I am sorry for the delay in getting back to you with a decision, but I have now received the full set of referee reports.

As you can see from the comments below, the study received a mixed response. While referee #1 is not convinced that the study adds enough new insight over recent work, referees #2 and 3 are more supportive. I have carefully looked at the referees' comments and the issues raised and I do find that the study adds important insight. However, it is also clear that more experiments are needed for consideration here. Given this, I would therefore like to invite you to submit a revised manuscript that addresses the concerns raised by referees #2 and 3. I should add that it is EMBO Journal policy to allow only a single major round of revision only and that it is therefore important to resolve the raised issues at this stage.

REFeree REPORTS.

Referee #1:

In this manuscript the authors showed that RNaseH2 is promotes L1 retrotransposition in transformed cell lines, using a reporter indicator cassette combined with biochemical evidence. While interesting, the data is not surprising since previous evidence has suggested such a link (Bartsch et al 2017). The authors' attempt to situate their findings in the context of AGS seems overstated. There are many genes implicated in the different subtypes of AGS. The idea that there is a unifying mechanism to explain all these subtypes is just a hypothesis (Volkman & Stetson 2014) and was never fully systematic tested using relevant cell lines in side-to-side experiments. This is also something the authors could have done here but didn't. As it is, this report is another evidence

of RNases during L1 retrotransposition.

Referee #2:

The authors nicely demonstrate that RNase H2 activity is required for efficient LINE-1 retrotransposition and propose a mechanism wherein RNase H2 degrades the RNA strand of the RNA-DNA hybrid generated during following the reverse transcription step of the retrotransposition pathway (TPRT). Because mutations in RNase H2 are causes of AGS, these results suggest that in contrast to other AGS-associated proteins that act as LINE-1 repressors, RNase H2 promotes LINE-1 transposition. The authors suggest that LINE-1 retroelement RNA-cDNA intermediates may be drivers of AGS pathology.

The manuscript is clearly written, and the data presented is convincing. In order to make a direct connection between the role of RNase H2 in promoting LINE-1 transposition and AGS pathology, the authors could perform some additional experiments discussed below.

Specific comments:

Pages 10-11, discussion of Figure 5A, B:

Please comment on the fact that although the activity of the RNase H2-SoF mutant on a RNA:DNA heteroduplex is significantly reduced *in vitro*, within a cell RNase H1 is also present and proficient at cleaving this type of substrate.

Figure 5A:

The altered cleavage pattern produced by cleavage by the SoF mutant is intriguing. It would also be interesting to test the cleavage efficiencies and patterns for one or more of the RNase H2-AGS mutants.

Figure 5.

In order to make a direct connection to AGS, my suggestion is to perform this experiment using one or more of the AGS mutants of RNase H2. This would more convincingly establish a connection between reduced LINE-1 retrotransposition, RNase H2 activity and disease.

Figure 6D.

When looking at this model, the role of RNase H1 in this pathway is confusing and could be removed from the figure and referenced in the figure legend as having a minor contribution.

Figure S8.

The authors clearly demonstrate, using this experimental approach, that loss of RNase H2 activity does not cause a significant increase in missense mutations or 2-5 bp deletions within the NEO cassette. Is there any possibility that larger genomic rearrangements may be occurring? It is difficult to analyze when looking at the gel in Figure S8C, but is there a difference in the PCR product size or amplification efficiency in the KO cell lines when compared to WT? Even if a difference is not apparent, the authors should mention the possibility that larger forms of genome instability may be triggered by knockdown of RNase H2 activity and/or alterations in LINE-1 retrotransposition.

Referee #3:

- general summary and opinion about the principle significance of the study, its questions and findings

Publication of two papers describing TREX1 and RNASEH2 mutations in Aicardi-Goutieres Syndrome patients led to a search for the relationship between patients' phenotypes and defects in these four genes (TREX1 and the three genes encoding the heterotrimeric RNase H2 protein). The first major paper addressed the relationship of the TREX1 function to AGS. Mice deleted for *Trex1* live only a few weeks after birth with death resulting from severe inflammatory myocarditis. *Trex1* is abundant in the cytoplasm. A substantial increase in abundance of ssDNAs were found in the cytoplasm of mouse hearts of *Trex1*^{-/-} mice. The increase in cytoplasmic ssDNA was derived to a

large extent from endogenous retroviral elements. The authors posited that activation of retroviruses provided the cytoplasmic ssDNA which in turn induced the innate immune response. A commentary on that paper presented an elaboration of that idea to include RNase H2 in the pathway whereby RNase H2 defects would result in incomplete removal of RNA of RNA/DNA hybrids derived from activated retroviruses leading to cytosolic stimulation of the innate immune response. Support for the activation of retroviruses hypothesis came when it was reported that the life of mice deleted for *Trex1* could be extended from a few weeks to more than a year by treatment with anti-retroviral drugs.

Directly asking if loss or decrease of RNase H2 activity activates retroviruses is confounded by the role of RNase H2 in resolving two distinct substrates: ribonucleotides incorporated in DNA during replication and repair and RNA/DNA hybrids formed during transcription or reverse transcription. Significant progress in understanding which of the two activities is related to a particular phenotype has been made in *Saccharomyces cerevisiae* by use of a separation of function mutation in which RNase H2 retains only the ability to resolve RNA/DNA hybrids.

Currently, mutations in seven genes have been identified in AGS patients, three of which encode the subunits of RNase H2. Three different mouse models, (i) null mutation in the gene encoding *Rnaseh2b* or deletion of the *Rnaseh2b* or *2c* gene and (ii) two hypomorphic mutations in the *Rnaseh2bA174T* (equivalent to the most frequently detected RNase H2 mutation in AGS patients *RNASEHA177T*), and *Rnaseh2aG37S* have been described. These models do not exactly recapitulate the human disorder but have provided important information about the role of RNase H2 in mammalian cells. Mice with null mutations result in early embryonic lethality with enormous numbers of ribonucleotides in genomic DNA, homozygous *Rnaseh2bA177T* mice live normal lives, and *Rnaseh2aG37S* homozygous mice are perinatal lethal. Induction of the innate immune system is the hallmark of AGS and both hypomorphic mutations have been shown to activate the DNA-dependent innate immune pathway. Recent evidence has shown chromosomal DNA damage in RNase H2-defective cells leads to formation of micronuclei which could provide DNA in the cytoplasm to activate the innate immune response. None of these mouse models of AGS are particularly informative as to which of the two activities is responsible for the observed phenotypes. Embryonic lethality of mice without RNase H2 has limited proper studies of the type reported in this manuscript. However, recent reports indicate that RNase H2 is not required for growth in several human cell lines. It is this resource that the authors of the manuscript have exploited to address the involvement of RNase H2 in endogenous retroviral activation. One publication cited in this manuscript (Bartsch et al) reported that Line-1 and Alu elements are not replicated in HeLa and HEK293 cells with *RNASEH2AKO* cells, indicating a role for RNase H2 in transposition of these elements. But, rather than finding loss of RNase H2 activating these viruses, they reported that RNase H2 is necessary for activation. In the manuscript under review here it seems that the authors have independently uncovered the loss of retrotransposition of Line-1 in *RNASEH2A* KO cell line and chose to describe in more detail the process in which RNase H2 is involved. Their data do confirm that RNase H2 is indeed required for activation of Line-1 elements, a retroelement with an RNase H activity.

The authors have shown that the absence of *RNASEH2A* leads to loss of Line-1 retrotransposition in three different human cell lines, that a xenotropic Line-1-like retroelement from Zebra Fish responds in HeLa cells with *RNASEHAKO* identically as if it were a mammalian retroelement, and that endogenous retroelements whose reverse transcriptase includes an RNase H activity are unaffected by the absence of RNase H2.

When knocking out genes in cell lines there is always a possibility of modifying a gene other than that targeted creating undesirable and/or inexplicable results. The data shown in this manuscript addressed this issue by creating multiple KO lines and even employed control cell lines which were not defective in RNase H2 but had been generated as byproducts of the search for authentic *RNase H2A* KO lines. By supplying *RNASEH2*-expression systems to the KO cell line (HeLa) they found they could restore normal levels retrotransposition to the KO cell lines. They extended their findings to two other cell lines, indicating that loss retrotransposition and restoration of retrotransposition is not limited to a single cell type. They also introduced a zebra fish line-2 (Line-1-like) element into HeLa cells and found this xenotropic retroelement responded to RNase H2 presence or absence as the human elements.

The authors asked if the human version of the separation of function mutant form of *S. cerevisiae* *SoF-RNase H2* could restore retrotransposition in their HeLa *RNASEH2AKO* cell line. This is the

first published description human SoF-RNase H2 and these authors suggest that, in addition to loss of incision at rNMPs in DNA, there are other defects. They find an altered cleavage pattern on their 18 bp labelled substrate and lower RNase H2 hybrid activity. These two differences in enzymatic activity are attributed by the authors for the failure to restore retrotransposition in the HeLa RNASEH2A KO cells. The residual RNase H2 hybrid activity is <50% of WT. The authors propose that the SoF-RNase H2 is insufficiently active to replace the WT RNase H2 activity.

However, the authors suggested the possibility that rNMPs in the "blasticidin" gene might be particularly susceptible to mutagenesis in the absence of RNase H2, in particular the RER activity. The hypothetical inactivation would appear as if loss retrotransposition had occurred since read out of the assay was resistance to blasticidin. The authors describe published results that reverse transcriptase of Line-1 elements does incorporate rNMPs during replication. To test this possibility, they allowed HeLa RNASEH2A KO cells to grow for two and five days in the absence of blasticidin during which time mutations in the blasticidin gene could accumulate. They sequenced PCR fragments from 50 clones, only 13 of which had single point mutations and found no 2-5 bp deletions which are found uniquely when rNMPs are present in DNA. These data are consistent with a low frequency of loss of the blasticidin function - a level not high enough to affect their retrotransposition assays. They examined too few clones to report as a mutation rate (as indicated in Figure S8) and possibly find any 2-5 bp deletions. This latter type of mutation occurs primarily in some but not all short repeats in DNA sequence. Does blasticidin have such repeats in the interval examined? Although the mutagenic test was not performed when SoF-RNase H2 was in RNASE2KO cells, it is possible, by inference, to conclude embedded rNMP-induced mutagenesis in the presence of SoF-RNase H2. However, deletions and hypomorphic mutations do always generate identical phenotypes.

- specific major concerns essential to be addressed to support the conclusions

The interpretation that the SoF-RNase H2 is defective is based on very little evidence. The authors found that overexpressing RNase H2 increases retrotransposition above that in RNase H2 WT cells. High level expression of RNase H1 also partially restores retrotransposition in RNASEH2A KO cells, indicating that the amount of Hybrid activity is important. Any RNase H activity above a threshold level may restore retrotransposition. The approach of overexpression provides an avenue to raise the total activity of SoF-RNase H2 above the threshold level by increasing activity from <50% to a much higher level. The possibility of partial restoration of retrotransposition by SoF-RNase H2 needs to be tested. If the increased level of SoF-RNase H2 Hybrid activity fails to achieve an increase retrotransposition, the absence RER activity in the SoF-RNase H2 becomes more important.

The model presented in Figure 6D appears to me to indicate that RNA is removed prior to the second strand break to allow the completion of the insertion of the Line-1 element. There is increasing evidence that DNA break repair can be mediated by RNA (see

<https://www.nature.com/articles/nature13682>

<http://www.sciencedirect.com/science/article/pii/S0092867416313824>

<https://www.nature.com/articles/ncb3643>).

Were the RNA/DNA hybrid to remain present after the second strand break, the RNA of the hybrid would provide a target for RNase H2 to bind and attract the DNA repair machinery via its PCNA binding peptide for replacing the RNA with DNA. RER activity could be useful in this process.

I strongly request that reference of the two activities as Type1 and Type2 be changed to the more transparent and descriptive terms - Hybrid and RER.

- minor concerns that should be addressed

I found it difficult to easily distinguish when the antibodies to RNase H2 and RNase H2A were being used. I am not sure how to make this better.

Figures in text are frequently not in numerical order in the text.

Target PRIMERD typo on Results third page after heading

Is the statement on the third page of Results "To our knowledge..." correct considering the Bartsch paper?

The figure legends often have no description of symbols or notations such as * nucleic acids filled or open circles

RESPONSE TO REFEREES' COMMENTS ON "RNase H2, mutated in Aicardi-Goutières syndrome, promotes LINE-1 retrotransposition", EMBOJ-2017-98506.

We would like to take this opportunity to thank the reviewers for their interest in our work and their detailed and insightful comments. These have been invaluable to us in further strengthening our conclusion that the activity of RNase H2 on RNA:DNA substrate is important for efficient LINE-1 retrotransposition, and also in enhancing the relevance of our findings to AGS.

In particular, in the modified manuscript,

- We provide a more in-depth analysis of the RNase H2 Separation of Function mutant, showing that it has substantially reduced substrate affinity. In spite of this, and consistent with the role of RNase H2 activity in the LINE-1 RNA:cDNA heteroduplex, its overexpression supports increased LINE-1 retrotransposition.
- We have analysed RNASEH2A-KO cells genetically complemented with RNASEH2A carrying AGS missense mutations, and show that these have significantly reduced LINE-1 retrotransposition. This strongly suggests that AGS patients with RNase H2 mutations are likely to have reduced levels of LINE-1 retrotransposition.

Below, we provide a detailed point-by-point response to all specific questions raised by the reviewers. Reviewers' comments are italicized (grey); bullet points precede our responses. To assist the reviewers, new text in the revised manuscript is marked in blue.

Point-by-point response

Referee #2:

The authors nicely demonstrate that RNase H2 activity is required for efficient LINE-1 retrotransposition and propose a mechanism wherein RNase H2 degrades the RNA strand of the RNA-DNA hybrid generated during following the reverse transcription step of the retrotransposition pathway (TPRT). Because mutations in RNase H2 are causes of AGS, these results suggest that in contrast to other AGS-associated proteins that act as LINE-1 repressors, RNase H2 promotes LINE-1 transposition. The authors suggest that LINE-1 retroelement RNA-cDNA intermediates may be drivers of AGS pathology.

The manuscript is clearly written, and the data presented is convincing. In order to make a direct connection between the role of RNase H2 in promoting LINE-1 transposition and AGS pathology, the authors could perform some additional experiments discussed below.

- We thank the Referee for his/her supportive comments and for encouraging us to perform additional experiments to strengthen the

link between the role for RNase H2 in LINE-1 retrotransposition and AGS pathology.

Specific comments:

Pages 10-11, discussion of Figure 5A, B:

Please comment on the fact that although the activity of the RNase H2-SoF mutant on a RNA:DNA heteroduplex is significantly reduced in vitro, within a cell RNase H1 is also present and proficient at cleaving this type of substrate.

- We agree that it is important to state this clearly, and we now include the following sentence in the revised manuscript, when discussing complementation of the RNASEH2A-KO with SoF-RNASEH2A.

“RNase H1 is expressed in all of these cells, explaining the residual activity against RNA:DNA substrate in RNASEH2A-KO cells complemented with empty vector or RNASEH2A-CD.”

- In addition, in the figure legend of Fig EV4 we state:

“Because RNase H1 is expressed in all of these cells, activity measured against RNA:DNA heteroduplex substrate in RNASEH2A-KO cell lysates is not completely absent. In addition, it is likely that other nucleases present in the cell lysate act (non-specifically) on the substrate, causing further background activity on both substrates.”

- Finally, in the original manuscript, we already noted the contribution of RNase H1 to cellular nuclease activity against RNA:DNA heteroduplexes. We have made edits to this section that, we hope, will highlight this more clearly.

“We assume RNase H1 to be expressed at normal levels in RNASEH2A-KO cells, and because of the marked reduction in LINE-1 retrotransposition in such null cells, it is unlikely that RNase H1 plays a major role in L1 retrotransposition. However, its activity against RNA:DNA hybrids may still contribute and could explain the variable level of remaining retrotransposition observed in our different RNase H2 null cells; retrotransposition was reduced between 4- and 15-fold compared to parental cells, depending on the cell line used. We therefore tested the effect of complementing HeLa RNASEH2A-KO cells with RNase H1.”

Figure 5A:

The altered cleavage pattern produced by cleavage by the SoF mutant is intriguing. It would also be interesting to test the cleavage efficiencies and patterns for one or more of the RNase H2-AGS mutants.

- As suggested, we have now performed this assay with purified recombinant RNase H2 complexes containing AGS mutations: RNASEH2A-G37S and E225G, as well as RNASEH2B-A177T and RNASEH2C-R69W (**Fig EV5**). As we have previously shown (Reijns et al, 2011), the A177T mutation had limited impact on enzyme activity. On the other hand, the G37S, E225G and R69W mutations all caused a substantial reduction in RNase H2 activity. This is consistent with impaired RNase H2 function in AGS patient cells with mutations in any of the three RNase H2 subunits. However, none of the AGS mutant enzymes displayed the change in cleavage pattern observed for the SoF mutant. We speculate that the changed cleavage pattern for the latter may be due to its reduced substrate affinity (see our response to Referee 3), an effect that may be more pronounced closer to the substrate 3' end.
- We describe these results in a new section of the revised manuscript, "**Reduced LINE-1 retrotransposition due to RNase H2 disease mutations**" (see below).

Figure 5.

In order to make a direct connection to AGS, my suggestion is to perform this experiment using one or more of the AGS mutants of RNase H2. This would more convincingly establish a connection between reduced LINE-1 retrotransposition, RNase H2 activity and disease.

- We thank the Referee for suggesting this important experiment. We complemented RNASEH2A-KO cells with AGS mutant RNASEH2A. For this, we chose G37S and E225G, the only two missense mutations that have been found as causative homozygous changes in the RNASEH2A subunit in AGS patients (Rice et al, 2013). Thus, the RNase H2 status of these complemented cells most closely resembles that of AGS patient cells.
- We show that these cells express mutant RNASEH2A (**Fig 7A**), leading to a small increase in cellular RNase H2 activity (**Fig 7B**). We next analysed LINE-1 retrotransposition (**Fig 7C-D**) and found a partial rescue of retrotransposition rates in the complemented cells. In the revised manuscript, we describe these results as follows:

“Reduced LINE-1 retrotransposition due to RNase H2 disease mutations

Overall, our data suggest that LINE-1 retrotransposition is likely to be reduced in cells from AGS patients with RNase H2 mutations, as most of these are known to cause reduced cellular enzyme activity. We confirmed the impact of disease mutations in any of the three subunits using recombinant RNase H2, but did not find the same change in cleavage pattern observed for the separation of function mutant (Fig EV5A, B). We speculate that the altered cleavage pattern for the latter may be due to its reduced substrate affinity, an effect that may be more pronounced closer to the substrate 3' end. To determine the cellular effect of RNase H2 disease mutations, we complemented RNASEH2A-KO cells with RNASEH2A-G37S and RNASEH2A-E225G, the only two missense mutations that have been found so far as causative homozygous changes in the catalytic subunit in AGS patients (Rice et al, 2013). Western-blot analyses revealed that these cells express mutant RNASEH2A (Fig 7A), leading to a small, but significant increase in cellular RNase H2 activity (Fig 7B). Notably, we observed partial rescue of LINE-1 retrotransposition in cells complemented with RNASEH2A-G37S and RNASEH2A-E225G (Fig 7C, D). Importantly, both AGS mutants displayed significantly reduced retrotransposition compared to RNASEH2A-WT complemented cells ($n = 3$, $P = 0.003$ for G37S, and $P = 0.018$ for E225G). Therefore, based on our data, we would expect AGS patients with RNase H2 mutations to have reduced levels of productive LINE-1 retrotransposition.”

Figure 6D.

When looking at this model, the role of RNase H1 in this pathway is confusing and could be removed from the figure and referenced in the figure legend as having a minor contribution.

- We agree, and have removed RNase H1 from the revised model cartoon, referring to it only in the figure legend.

Figure S8.

The authors clearly demonstrate, using this experimental approach, that loss of RNase H2 activity does not cause a significant increase in missense mutations or 2-5 bp deletions within the NEO cassette. Is there any possibility that larger genomic rearrangements may be occurring? It is difficult to analyze when looking at the gel in Figure S8C, but is there a difference in the PCR product size or amplification efficiency in the KO cell

lines when compared to WT? Even if a difference is not apparent, the authors should mention the possibility that larger forms of genome instability may be triggered by knockdown of RNase H2 activity and/or alterations in LINE-1 retrotransposition.

- We agree with the Referee that it is formally possible that such genome instability impacts on LINE-1 retrotransposition in RNASEH2A-KO cells. However, the intron-based PCR assay is not designed to detect larger rearrangements, and we found no evidence for differently sized PCR products or differential PCR amplification efficiency. In addition, it seems unlikely to us that genome rearrangements play a major part in the reduced LINE-1 retrotransposition in RNase H2 null cells, as it is difficult to envisage how this would explain the increase in retrotransposition observed upon RNase H2 overexpression.
- To acknowledge the possible interplay between genomic rearrangements and LINE-1 retrotransposition, we now include the following in the Discussion of the revised manuscript:

“Furthermore, RNase H2 deficiency can also cause larger genomic rearrangements (Reijns et al, 2012) which might impact on LINE-1 retrotransposition. Although we cannot rule out that this type of genome instability interferes with retrotransposition, or even that non-productive retrotransposition contributes to increased genomic rearrangements in RNase H2 null cells, it seems unlikely that this is the reason for the reduced LINE-1 retrotransposition we observe, particularly as RNase H2 overexpression leads to increased rates of retrotransposition.”

Referee #3:

- general summary and opinion about the principle significance of the study, its questions and findings

Publication of two papers describing TREX1 and RNASEH2 mutations in Aicardi-Goutieres Syndrome patients led to a search for the relationship between patients' phenotypes and defects in these four genes (TREX1 and the three genes encoding the heterotrimeric RNase H2 protein). The first major paper addressed the relationship of the TREX1 function to AGS. Mice deleted for Trex1 live only a few weeks after birth with death resulting from severe inflammatory myocarditis. Trex1 is abundant in the cytoplasm. A

substantial increase in abundance of ssDNAs were found in the cytoplasm of mouse hearts of Trex1^{-/-} mice. The increase in cytoplasmic ssDNA was derived to a large extent from endogenous retroviral elements. The authors posited that activation of retroviruses provided the cytoplasmic ssDNA which in turn induced the innate immune response. A commentary on that paper presented an elaboration of that idea to include RNase H2 in the pathway whereby RNase H2 defects would result in incomplete removal of RNA of RNA/DNA hybrids derived from activated retroviruses leading to cytosolic stimulation of the innate immune response. Support for the activation of retroviruses hypothesis came when it was reported that the life of mice deleted for Trex1 could be extended from a few weeks to more than a year by treatment with anti-retroviral drugs.

*Directly asking if loss or decrease of RNase H2 activity activates retroviruses is confounded by the role of RNase H2 in resolving two distinct substrates: ribonucleotides incorporated in DNA during replication and repair and RNA/DNA hybrids formed during transcription or reverse transcription. Significant progress in understanding which of the two activities is related to a particular phenotype has been made in *Saccharomyces cerevisiae* by use of a separation of function mutation in which RNase H2 retains only the ability to resolve RNA/DNA hybrids.*

Currently, mutations in seven genes have been identified in AGS patients, three of which encode the subunits of RNase H2. Three different mouse models, (i) null mutation in the gene encoding Rnaseh2b or deletion of the Rnaseh2b or 2c gene and (ii) two hypomorphic mutations in the Rnaseh2bA174T (equivalent to the most frequently detected RNase H2 mutation in AGS patients RNASEHA177T), and Rnaseh2aG37S have been described. These models do not exactly recapitulate the human disorder but have provided important information about the role of RNase H2 in mammalian cells. Mice with null mutations result in early embryonic lethality with enormous numbers of ribonucleotides in genomic DNA, homozygous Rnaseh2bA177T mice live normal lives, and Rnaseh2aG37S homozygous mice are perinatal lethal. Induction of the innate immune system is the hallmark of AGS and both hypomorphic mutations have been shown to activate the DNA-dependent innate immune pathway. Recent evidence has shown chromosomal DNA damage in RNase H2-defective cells leads to formation of micronuclei which could provide DNA in the cytoplasm to activate the innate immune response. None of these mouse models of AGS are particularly informative as to which of the two activities is responsible for the observed phenotypes.

Embryonic lethality of mice without RNase H2 has limited proper studies of the type reported in this manuscript. However, recent reports indicate that RNase H2 is not required for growth in several human cell lines. It is this resource that the authors of the manuscript have exploited to address the involvement of RNase H2 in endogenous retroviral activation. One publication cited in this manuscript (Bartsch et al) reported that Line-1 and Alu elements are not replicated in HeLa and HEK293 cells with RNASEH2AKO cells, indicating a role for RNase H2 in transposition of

these elements. But, rather than finding loss of RNase H2 activating these viruses, they reported that RNase H2 is necessary for activation. In the manuscript under review here it seems that the authors have independently uncovered the loss of retrotransposition of Line-1 in RNASEH2A KO cell line and chose to describe in more detail the process in which RNase H2 is involved. Their data do confirm that RNase H2 is indeed required for activation of Line-1 elements, a retroelement with an RNase H activity.

The authors have shown that the absence of RNASEH2A leads to loss of Line-1 retrotransposition in three different human cell lines, that a xenotropic Line-1-like retroelement from Zebra Fish responds in HeLa cells with RNASEHAKO identically as if it were a mammalian retroelement, and that endogenous retroelements whose reverse transcriptase includes an RNase H activity are unaffected by the absence of RNase H2.

When knocking out genes in cell lines there is always a possibility of modifying a gene other than that targeted creating undesirable and/or inexplicable results. The data shown in this manuscript addressed this issue by creating multiple KO lines and even employed control cell lines which were not defective in RNase H2 but had been generated as byproducts of the search for authentic RNase H2A KO lines. By supplying RNASEH2-expression systems to the KO cell line (HeLa) they found they could restore normal levels retrotransposition to the KO cell lines. They extended their findings to two other cell lines, indicating that loss retrotransposition and restoration of retrotransposition is not limited to a single cell type. They also introduced a zebra fish line-2 (Line-1-like) element into HeLa cells and found this xenotropic retroelement responded to RNase H2 presence or absence as the human elements.

*The authors asked if the human version of the separation of function mutant form of *S. cerevisiae* SoF-RNase H2 could restore retrotransposition in their HeLa RNASEH2AKO cell line. This is the first published description human SoF-RNase H2 and these authors suggest that, in addition to loss of incision at rNMPs in DNA, there are other defects. They find an altered cleavage pattern on their 18 bp labelled substrate and lower RNase H2 hybrid activity. These two differences in enzymatic activity are attributed by the authors for the failure to restore retrotransposition in the HeLa RNASEH2A KO cells. The residual RNase H2 hybrid activity is <50% of WT. The authors propose that the SoF-RNase H2 is insufficiently active to replace the WT RNase H2 activity.*

- We thank the Referee for his/her detailed analysis and supportive comments.

However, the authors suggested the possibility that rNMPs in the "blasticidin" gene might be particularly susceptible to mutagenesis in the absence of RNase H2, in particular the RER activity. The hypothetical inactivation would appear as if loss retrotransposition had occurred since read out of the assay was resistance to blasticidin. The authors describe

published results that reverse transcriptase of Line-1 elements does incorporate rNMPs during replication. To test this possibility, they allowed HeLa RNASEH2A KO cells to grow for two and five days in the absence of blasticidin during which time mutations in the blasticidin gene could accumulate. They sequenced PCR fragments from 50 clones, only 13 of which had single point mutations and found no 2-5 bp deletions which are found uniquely when rNMPs are present in DNA. These data are consistent with a low frequency of loss of the blasticidin function - a level not high enough to affect their retrotransposition assays. They examined too few clones to report as a mutation rate (as indicated in Figure S8) and possibly find any 2-5 bp deletions. This latter type of mutation occurs primarily in some but not all short repeats in DNA sequence. Does blasticidin have such repeats in the interval examined? Although the mutagenic test was not performed when SoF-RNase H2 was in RNASE2KO cells, it is possible, by inference, to conclude embedded rNMP-induced mutagenesis in the presence of SoF-RNase H2. However, deletions and hypomorphic mutations do always generate identical phenotypes.

- The Referee is correct and the purpose of these experiments, in which we sequenced part of the spliced neomycin resistance gene, was to rule out increased mutagenesis in RNase H2 null cells as the cause of the observed reduction in neomycin resistant colonies after LINE-1 retrotransposition. We are pleased that the Referee agrees with our conclusion that differential rates of mutagenesis do not underlie the lower number of colonies for RNASEH2A-KO cells. To increase the strength of these observations, we expanded the analysis by sequencing additional clones (total, n=136). Consistent with our previous findings, we observed a similar percentage of mutated sequences in RNase H2 proficient and deficient cells (**Appendix Figure S3E**). In addition, we analysed the mutation frequency in RNASEH2A-KO cells complemented with SoF RNASEH2A (33 mutated out of 52 sequenced, 63%) and found that it was similar to cells complemented with EV (17/35, 49%) or WT RNASEH2A (23/49, 47%).
- The Referee makes an important point about the occurrence of 2-5 bp deletions: in RNase H2 null *S. cerevisiae* they were shown to occur in short tandem repeats in particular sequence contexts. It is unknown whether this mutational signature is similarly associated with RER deficiency in mammalian cells or whether it could occur during LINE-1 retrotransposition. Nonetheless, to give a better idea of the likelihood of finding similar deletions, we compared the occurrence of such repeats in our reporter constructs (neomycin and blasticidin) to those described in the yeast literature.

- We analysed the nucleotide context of short deletions found in RNase H2 null yeast strains reported for three different reporter genes: URA3 (Clark et al, 2011), CAN1 (Kim et al, 2011; Potenski et al, 2014) and LYS2 (Kim et al, 2013). Out of the 299 2-5 bp deletions documented by these studies, 247 (83%) occurred in perfect tandem repeats, and 236 were 2 bp long (79%). Of these 2 bp deletions, the majority (83%) occurred at CA:TC and GA:TC repeats. To put this in context of the frequency at which tandem repeats occur in the reporters, 21% of the URA3 coding sequence is composed of tandem repeats, and 58% of these are CA:TC or GA:TC dinucleotide repeats. The section of CAN1 used in these studies is composed of 21.5% dinucleotide repeats, with 46% of these CA:TC or GA:TC dinucleotide repeats.
- In the retrotransposition indicator cassettes used in this study, tandem dinucleotide repeats are present in the PCR amplicon of the spliced Neomycin resistance gene (15% of this sequence is composed of tandem repeats, with 56% of these at CA:TC or GA:TC repeats), as well as the Blasticidin resistance gene (19.4% of this sequence is composed of tandem repeats, with 49% of these at CA:TC or GA:TC repeats). Thus, the frequencies of tandem dinucleotide repeats in our reporters are only slightly below those found in the yeast reporters. To highlight the presence and position of tandem dinucleotide repeats, we updated the scheme of the neomycin indicator cassette (Appendix Figure S3D). In addition we've added the following to the Results section:

“These mutations are most likely to occur at tandem dinucleotide repeats, particularly CA:TC and GA:TC (Clark et al, 2011; Kim et al, 2013; Kim et al, 2011; Potenski et al, 2014). These repeats occur in our retrotransposition reporters at rates similar to the yeast reporters”

- Altogether, this additional work helped to strengthen our previous conclusion: the LINE-1 reporters in RER deficient cells do not accumulate more mutations than in RER proficient cells.

- specific major concerns essential to be addressed to support the conclusions

The interpretation that the SoF-RNase H2 is defective is based on very little evidence.

The authors found that overexpressing RNase H2 increases

retrotransposition above that in RNase H2 WT cells. High level expression of RNase H1 also partially restores retrotransposition in RNASEH2A KO cells, indicating that the amount of Hybrid activity is important. Any RNase H activity above a threshold level may restore retrotransposition. The approach of overexpression provides an avenue to raise the total activity of SoF-RNase H2 above the threshold level by increasing activity from <50% to a much higher level. The possibility of partial restoration of retrotransposition by SoF-RNase H2 needs to be tested. If the increased level of SoF-RNase H2 Hybrid activity fails to achieve an increase retrotransposition, the absence RER activity in the SoF-RNase H2 becomes more important.

- We thank the Referee for proposing this experiment. As suggested, we determined the effect of overexpressing SoF-RNase H2, and find that it indeed leads to increased LINE-1 retrotransposition (**Fig 6A**). This is in agreement with a role for cellular RNase H activity acting on RNA:DNA heteroduplexes during retrotransposition.
- In addition, we measured enzyme kinetics for the SoF mutant enzyme and find that it has much reduced substrate affinity (**Fig EV4I**). As the previous enzyme activity assays were performed at a substrate concentration of 2 μM , a high concentration (at which the activity for SoF-RNase H2 is ~50% of wildtype), but well below the K_m^{SoF} of 3.8 μM , we speculate that the effective enzyme activity will be more strongly affected at lower, more physiologically relevant substrate concentrations. The reduced substrate affinity of SoF RNase H2 therefore provides a likely explanation for this mutant failing to rescue retrotransposition in complemented RNASEH2A-KO cells.
- In the revised manuscript, we describe these results as follows:

SoF RNase H2 overexpression supports increased LINE-1 retrotransposition, despite reduced substrate affinity

*We reasoned that overexpression of the RNase H2 SoF mutant may compensate for its reduced activity against RNA:DNA hybrids, and tested the effect of simultaneous overexpression of RNASEH2A-P40D/Y210A, RNASEH2B and RNASEH2C on LINE-1 retrotransposition. We found that overexpression of SoF RNase H2 indeed leads to increased LINE-1 retrotransposition compared to the β -arrestin control (**Fig 6A**). To further investigate why the separation of function mutant failed to rescue retrotransposition in the complemented RNASEH2A-KO cells, whereas its overexpression did support a higher rate of retrotransposition, we compared enzyme kinetics for SoF and wildtype RNase H2 on RNA:DNA (**Fig EV4I**).*

We established that the SoF mutant has much reduced substrate affinity ($K_m^{\text{SoF}} \sim 16x K_m^{\text{WT}}$), whereas its maximum substrate conversion rate is similar to that of WT RNase H2 ($k_{\text{cat}}^{\text{SoF}} \sim 0.83x k_{\text{cat}}^{\text{WT}}$). The reduced substrate affinity of SoF RNase H2 therefore provides a likely explanation for our observations. These findings are in keeping with a role for the activity of RNase H2 against RNA:DNA hybrids to support LINE-1 retrotransposition.

The model presented in Figure 6D appears to me to indicate that RNA is removed prior to the second strand break to allow the completion of the insertion of the Line-1 element. There is increasing evidence that DNA break repair can be mediated by RNA (see <https://www.nature.com/articles/nature13682> <http://www.sciencedirect.com/science/article/pii/S0092867416313824> <https://www.nature.com/articles/ncb3643>).

Were the RNA/DNA hybrid to remain present after the second strand break, the RNA of the hybrid would provide a target for RNase H2 to bind and attract the DNA repair machinery via its PCNA binding peptide for replacing the RNA with DNA. RER activity could be useful in this process.

- The precise molecular events underlying LINE-1 retrotransposition are not fully understood. However, the most widely accepted model in the LINE-1 field indeed suggests that second-strand DNA cleavage occurs after completion of reverse transcription, preventing the generation of inverted-deleted LINE-1s by a mechanism known as twin priming (Ostertag & Kazazian, 2001). As the Referee suggests, a more direct role for RNA:DNA hybrids in the LINE-1 retrotransposition process, akin to the role RNA is now thought to play in DSB repair, is therefore an intriguing possibility.
- To acknowledge the possibility that RNase H2 may promote LINE-1 retrotransposition in ways other than the model we propose, we now include the following text in the discussion of our revised manuscript:

“Our work shows that RNase H activity directed against RNA:DNA hybrids, mainly provided by cellular RNase H2, is important for efficient and productive LINE-1 retrotransposition. We interpret this to mean that it is involved in degrading LINE-1 RNA in the RNA:cDNA hybrid, allowing second strand synthesis and ultimately insertion into the genome. Alternative explanations are of course possible. As there is increasing evidence that DNA double strand break repair can be mediated by RNA (Keskin et al, 2014; Michelini et al, 2017; Ohle et al, 2016), one intriguing possibility is that RNA:DNA hybrids play a more active role in the LINE-1 retrotransposition process, for example by recruiting RNase H2 and DNA repair machinery.”

I strongly request that reference of the two activities as Type1 and Type2 be changed to the more transparent and descriptive terms - Hybrid and RER.

- As suggested, and to be more descriptive for those less familiar with RNase H enzymes, we have changed the use of Type 1 activity to “activity against RNA:DNA heteroduplexes” and Type 2 activity to “activity against single embedded ribonucleotides”.

- minor concerns that should be addressed

I found it difficult to easily distinguish when the antibodies to RNase H2 and RNase H2A were being used. I am not sure how to make this better.

- We initially used an antibody from Origene (TA306706) to successfully detect RNASEH2A. However, a later lot of this antibody (lot 8291-1404) performed poorly (with strong, non-specific bands close to the RNASEH2A band). Also, we found this anti-peptide antibody to detect RNASEH2A-P40D/Y210A less well than wildtype or D34A/D169A RNase H2, suggesting that the Tyr210 residue may be an important part of the epitope. For these reasons, we originally presented some of our blots with RNASEH2A detected using the anti-RNase H2 antibody.
- During revision we tested other anti-RNASEH2A antibodies and found a good alternative (Santa Cruz, sc-515475). We re-probed some of our blots with this new antibody, so that all RNASEH2A immunoblots in the revised manuscript now use antibodies that specifically detect this subunit. The anti-RNase H2 antibody was used throughout to detect the RNASEH2B and C subunits.

Figures in text are frequently not in numerical order in the text.

- In the revised manuscript, we did our utmost to ensure we refer to figures throughout the main text in the correct numerical order.

Target PRIMERD typo on Results third page after heading

- We have corrected this typo.

Is the statement on the third page of Results "To our knowledge..." correct considering the Bartsch paper?

- We thank the Referee for pointing this out. We have now changed this to:

“This would make RNase H2 an integral part of the LINE retrotransposition machinery.”

- Similarly, we have changed part of the final paragraph to:

“In summary, our work contributes to the mechanistic understanding of LINE-1 retrotransposition, as we demonstrate that cellular RNase H2 plays an integral part in LINE retrotransposition, explaining how LINE elements lacking an RNase H domain can retrotranspose.”

*The figure legends often have no description of symbols or notations such as * nucleic acids filled or open circles*

- We thank the Referee for this suggestion. However, we simply followed the EMBO Press Figure Guidelines, which state: *“Try to avoid verbal explanations (for example, ‘broken line’ or ‘filled black triangles’) in the written figure legend / caption itself. Place visual cues – a figure key listing the icons and their meaning – in the figure itself.”* Ultimately, if the manuscript is accepted for publication, we will take advice from the EMBO Journal copy editors to ensure figures and legends are clear to the reader.

Rebuttal References

Clark AB, Lujan SA, Kissling GE, Kunkel TA (2011) Mismatch repair-independent tandem repeat sequence instability resulting from ribonucleotide incorporation by DNA polymerase epsilon. *DNA repair* **10**: 476-482

Keskin H, Shen Y, Huang F, Patel M, Yang T, Ashley K, Mazin AV, Storici F (2014) Transcript-RNA-templated DNA recombination and repair. *Nature* **515**: 436-439

Kim N, Cho JE, Li YC, Jinks-Robertson S (2013) RNARatioDNA hybrids initiate quasi-palindrome-associated mutations in highly transcribed yeast DNA. *PLoS genetics* **9**: e1003924

Kim N, Huang SN, Williams JS, Li YC, Clark AB, Cho JE, Kunkel TA, Pommier Y, Jinks-Robertson S (2011) Mutagenic processing of ribonucleotides in DNA by yeast topoisomerase I. *Science* **332**: 1561-1564

Michellini F, Pitchiaya S, Vitelli V, Sharma S, Gioia U, Pessina F, Cabrini M, Wang Y, Capozzo I, Iannelli F, Matti V, Francia S, Shivashankar GV, Walter NG, d'Adda di Fagagna F (2017) Damage-induced lncRNAs control the DNA damage

response through interaction with DDRNAs at individual double-strand breaks. *Nature cell biology* **19**: 1400-1411

Ohle C, Tesorero R, Schermann G, Dobrev N, Sinning I, Fischer T (2016) Transient RNA-DNA Hybrids Are Required for Efficient Double-Strand Break Repair. *Cell* **167**: 1001-1013 e1007

Ostertag EM, Kazazian HH, Jr. (2001) Twin priming: a proposed mechanism for the creation of inversions in L1 retrotransposition. *Genome Res* **11**: 2059-2065

Potenski CJ, Niu H, Sung P, Klein HL (2014) Avoidance of ribonucleotide-induced mutations by RNase H2 and Srs2-Exo1 mechanisms. *Nature* **511**: 251-254

Reijns MA, Bubeck D, Gibson LC, Graham SC, Baillie GS, Jones EY, Jackson AP (2011) The structure of the human RNase H2 complex defines key interaction interfaces relevant to enzyme function and human disease. *J Biol Chem* **286**: 10530-10539

Reijns MA, Rabe B, Rigby RE, Mill P, Astell KR, Lettice LA, Boyle S, Leitch A, Keighren M, Kilanowski F, Devenney PS, Sexton D, Grimes G, Holt IJ, Hill RE, Taylor MS, Lawson KA, Dorin JR, Jackson AP (2012) Enzymatic removal of ribonucleotides from DNA is essential for mammalian genome integrity and development. *Cell* **149**: 1008-1022

Rice GI, Forte GM, Szykiewicz M, Chase DS, Aeby A, Abdel-Hamid MS, Ackroyd S, Allcock R, Bailey KM, Balottin U, Barnerias C, Bernard G, Bodemer C, Botella MP, Cereda C, Chandler KE, Dabydeen L, Dale RC, De Laet C, De Goede CG, Del Toro M, Effat L, Enamorado NN, Fazzi E, Gener B, Haldre M, Lin JP, Livingston JH, Lourenco CM, Marques W, Jr., Oades P, Peterson P, Rasmussen M, Roubertie A, Schmidt JL, Shalev SA, Simon R, Spiegel R, Swoboda KJ, Temtamy SA, Vassallo G, Vilain CN, Vogt J, Wermenbol V, Whitehouse WP, Soler D, Olivieri I, Orcesi S, Aglan MS, Zaki MS, Abdel-Salam GM, Vanderver A, Kisand K, Rozenberg F, Lebon P, Crow YJ (2013) Assessment of interferon-related biomarkers in Aicardi-Goutieres syndrome associated with mutations in TREX1, RNASEH2A, RNASEH2B, RNASEH2C, SAMHD1, and ADAR: a case-control study. *The Lancet Neurology* **12**: 1159-1169

Thanks for submitting your revised manuscript to The EMBO Journal. Your study has now been re-reviewed by referees #2 and 3 and their comments are provided below. As you can see both referees appreciate the introduced changes. I am therefore very pleased to accept the manuscript for publication here.

There are just a few minor things to resolve before I can send you the formal accept letter.

- Please respond to the remaining comment raised by referee #3. -----

REFEREE REPORTS.

Referee #2:

Having carefully read the authors' responses, and the resulting revisions to the manuscript, it is my opinion that this manuscript is acceptable for publication.

Referee #3:

The authors have responded to all of my concerns. The only one which has not been addressed is the "new" definition of RNase H1 and RNase H2. This is a semantic issue for which the authors add a definition that could and should be avoided.

This issue will be resolved as others provide new data.

Please see below for our response to the few minor remaining issues. Your comments are italicized (grey) and our comments are highlighted using blue text:

-1) Please respond to the remaining comment raised by referee #3.

[Reviewer 3 stated: "The authors have responded to all of my concerns. The only one which has not been addressed is the "new" definition of RNase H1 and RNase H2. This is a semantic issue for which the authors add a definition that could and should be avoided. This issue will be resolved as others provide new data."]

We believe the referee refers to the use of Type 1 and Type 2 RNase H activities, as in its original review this referee suggested the use of "Hybrid and RER" rather than Type 1 and Type 2 activities.

In response to the reviewers' original comment we changed Type 1 and Type 2 RNase H activity to "activity against RNA:DNA heteroduplexes" and "activity against single embedded ribonucleotides", respectively.

In our previous cover letter, we stated the reasons why we did not use the terms "hybrid and RER", suggested by the reviewer: we believe that using the suggested terms could be more rather than less confusing, and we indicated

this in our previous cover letter. This is because the term “Hybrid” is ambiguous, as it can be used to refer to RNA:DNA heteroduplexes but also to DNA containing embedded ribonucleotides. The term “RER” is not ambiguous; however, the term RER refers to the full pathway of ribonucleotide excision repair, in which cleavage by RNase H2 is just one of the steps in this pathway. In an updated version of our manuscript (uploaded as EMBOJ-2017-98506R1_Main-paper) we have ensured that the terms Type 1 and Type 2 RNase H activity are not used anywhere. In summary, we strongly feel that the way we now describe the two different types of RNase H activity is correct, non-ambiguous and should be easy to follow for non-specialists.

We hope this information is sufficient, but we would be happy to be guided by you if you feel more information is necessary.

Corresponding Author Name: Jose Luis Garcia-Perez

Journal Submitted to: EMBO J

Manuscript Number: EMBOJ-2017-98506